# Interpreting Representation Quality of DNNs for 3D Point Cloud Processing

**Wen Shen**[b*]    **Qihan Ren**[a]    **Dongrui Liu**[a]    **Quanshi Zhang**[a†]
[a]Shanghai Jiao Tong University    [b]Tongji University

## Abstract

In this paper, we evaluate the quality of knowledge representations encoded in deep neural networks (DNNs) for 3D point cloud processing. We propose a method to disentangle the overall model vulnerability into the sensitivity to the rotation, the translation, the scale, and local 3D structures. Besides, we also propose metrics to evaluate the spatial smoothness of encoding 3D structures, and the representation complexity of the DNN. Based on such analysis, experiments expose representation problems with classic DNNs, and explain the utility of the adversarial training. *The code will be released when this paper is accepted.*

## 1 Introduction

Deep neural networks (DNNs) have exhibited superior performance in various tasks, but the black-box nature of DNNs hampers the analysis of knowledge representations. Previous studies on explainable AI mainly focused on the following two directions. The first is to explain the knowledge encoded in a DNN, *e.g.* visualizing patterns encoded by a DNN [24, 36, 50] and estimating the saliency/importance/attribution of input variables *w.r.t.* the network output [20, 30, 57]. The second is to evaluate the representation power of a DNN, *e.g.* the generalization and robustness of feature representations.

This paper focuses on the intersection of the above two directions, *i.e.* analyzing the quality of knowledge representations of DNNs for 3D point cloud processing. Specifically, we aim to design metrics to illustrate properties of feature representations for different point cloud regions, including various types of regional sensitivities, the spatial smoothness of encoding 3D structures, and the representation complexity of a DNN. These metrics provide new perspectives to diagnose DNNs for 3D point clouds. For example, unlike the image processing relying on color information, the processing of 3D point clouds usually exclusively uses 3D structural information for classification. Therefore, a well-trained DNN for 3D point cloud processing is supposed to just use scale information and 3D structures for inference, and be robust to the rotation and translation.

**Regional sensitivities.** We first propose six metrics to disentangle the overall model vulnerability into the regional rotation sensitivity, the regional translation sensitivity, the regional scale sensitivity, and three types of regional structure sensitivity (sensitivity to edges, surfaces, and masses), so that we can use such sensitivity metrics to evaluate the representation quality of a DNN (as Fig. 1 (a-b) shows). Each sensitivity metric (let us take the rotation sensitivity for example) is defined as the vulnerability of the regional attribution when we rotate the 3D point cloud with different angles. The regional attribution is computed as the Shapley value, which is widely used as a unique unbiased estimation of an input variable's attribution [11, 20, 31, 37] from the perspective of game theory.

---

*This work was done when Wen Shen was an intern at Shanghai Jiao Tong University.

†Quanshi Zhang is the corresponding author. This study was done under the supervision of Dr. Quanshi Zhang. He is with the John Hopcroft Center and the MoE Key Lab of Artificial Intelligence, AI Institute, at the Shanghai Jiao Tong University, China.

35th Conference on Neural Information Processing Systems (NeurIPS 2021).

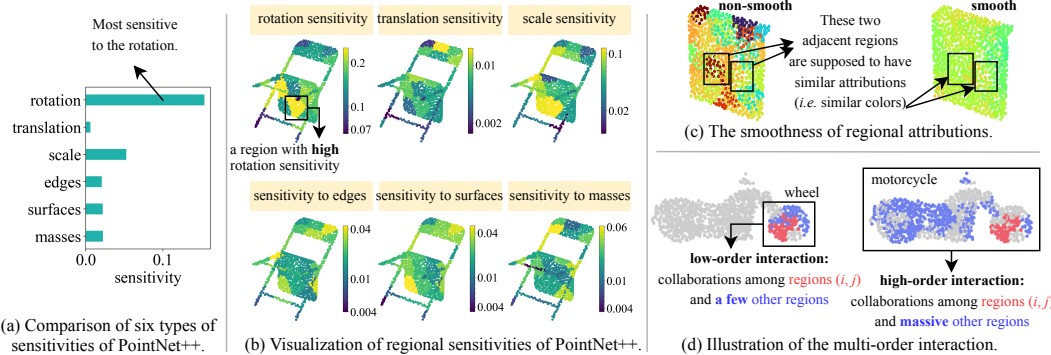

Figure 1: (a) Comparison of regional sensitivities of PointNet++ [28]. (b) Visualization of regional sensitivities of PointNet++ [28], where the heatmap is normalized in each sample. These colorbars are shown in log-scale. (c) Illustration of the regional smoothness. (d) Illustration of the representation complexity.

Based on the sensitivity metrics, we have discovered the following new insights, which have exposed problems with classic DNNs.

• *Insight 1.* Rotation robustness is the Achilles' heel of most DNNs for point cloud processing. The rotation sensitivities of PointNet [26], PointNet++ [28], DGCNN [43], the non-dynamic version of DGCNN (referred to as GCNN in this paper) [43], and PointConv [45] are much higher than other sensitivities (see Fig. 1 (a)).

• *Insight 2.* It is usually difficult for DNNs to extract rotation-robust features from 3D points at edges and corners. These points are usually vulnerable to rotation-based attacks.

• *Insight 3.* What's worse, DNNs usually cannot ignore features of such points at edges and corners. Instead, such rotation-sensitive points usually have large regional attributions.

• *Insight 4.* PointNet fails to model local 3D structures, because convolutional operations in PointNet encode each point independently. Thus, PointNet has low sensitivity to local 3D structures.

**Spatial smoothness.** Beyond the sensitivity metrics, we also evaluate the spatial smoothness of knowledge representations of DNNs. We define the spatial smoothness of knowledge representations as the similarity among the neighboring regions' attributions to network output. Most widely-used benchmark datasets for point cloud classification [46, 49] only contain objects with simple 3D structures (*e.g.* a bed mainly composed of a few surfaces in Fig. 1 (c)), in which adjacent regions usually have similar 3D structures. Therefore, most adjacent regions are supposed to have similar attributions to the network output. We also find that the adversarial training increases the spatial smoothness of knowledge representations.

**Representation complexity.** Besides regional sensitivities, we further extend the metric of multi-order interaction [53] to evaluate the representation complexity of a DNN, *i.e.* the maximum complexity of 3D structures that can be encoded in a DNN. The interaction of the $m$-th order measures the additional benefits brought by collaboration between two point cloud regions $i, j$ under the contexts of other $m$ regions, where $m$ reflects the contextual complexity of the collaboration between regions $i$ and $j$. High-order interactions usually represent global and complex 3D structures, corresponding to the collaboration between point cloud regions $i, j$ and massive other regions. Low-order interactions usually describe local and simple 3D structures, without being influenced by many contextual regions (see Fig. 1 (d)). Based on such interactions, we have discovered the following new insights.

• *Insight 5.* Most DNNs fail to encode high-order interactions. *I.e.* most DNNs mainly focus on local structures and do not extract many global and complex structures from normal samples.

• *Insight 6.* Rotation-sensitive regions usually activate abnormal high-order interactions (global and complex structures). The very limited global structures modeled by the DNN are usually over-fitted to training samples. Thus, when the sample is rotated, the activated global and complex structures usually appear as abnormal patterns.

• *Insight 7.* Besides, in terms of model robustness, adversarial training based on adversarial samples *w.r.t.* using rotations/translations for attack is a typical way to improve the robustness to rotation and translation. Thus, we use the proposed sensitivity metrics to explain the inner mechanism of how and why the adversarially trained DNN is robust to rotation and translation. We find that (1) the adversarial training shifts the attention from the global orientations and positions to the structural information, thereby boosting the sensitivity to local 3D structures. (2) The adversarial training usually increases both the quantity and the complexity of 3D structures (high-order interactions from normal samples) modeled in a DNN, which boosts the robustness to the rotation and translation.

## 2  Related work

**Deep learning on 3D point clouds:** Recently, a series of studies directly used DNNs to process 3D point clouds and have achieved promising performance in various 3D tasks [26, 51, 42, 13, 27, 47, 41, 18, 10, 32, 33]. PointNet [26] was the pioneer to use point-wise multi-layer perceptron to process point clouds and aggregate all individual point features into a global feature. To further extract information from local 3D structures, PointNet++ [28] recursively applied PointNet to capture hierarchical 3D structures. Likewise, Relation-Shape CNN [19] and PointCNN [18] also focused on hierarchical structures and improved the ability to extract contextual information. Due to the irregularity of 3D point clouds, some approaches considered the point set as a graph and further defined graph convolutions. DGCNN [43] dynamically constructed local graphs and conducted feature extraction via the EdgeConv operation. SPG [17] built a superpoint graph to process large-scale point clouds. GACNet [40] introduced an attention mechanism in graph convolutional networks. KCNet [34] proposed kernel correlation and graph pooling to aggregate contextual information. In addition, many approaches have been proposed to apply convolution operators to the point clouds. PointConv [45] and KPConv [39] utilized nonlinear functions to construct convolution weights from the input 3D coordinates. InterpCNN [23] interpolated features of 3D points to the discrete kernel weights coordinates, so as to implement the discrete convolution operator to point clouds. Instead of designing new architectures, our study focuses on the analysis of representation quality of existing classic DNNs for point cloud processing.

**Visualization or diagnosis of representations:** It is intuitive to interpret DNNs by visualizing the feature representations encoded in intermediate layers of DNNs [52, 35, 50, 22, 7], and estimating the pixel-wise saliency/importance/attribution of input variables *w.r.t* the network output [29, 15, 8, 57, 30, 5, 56]. Similarly, in the 3D domain, PointNet [26] visualized the subset of 3D points (namely the critical subset) that directly affected the network output. Additionally, Zheng *et al.* [55] specified the critical subset via building a gradient-based saliency map. In comparison, our study proposes to evaluate the representation quality of different point cloud regions.

**Quantitative evaluation of representations:** The quantitative evaluation of the representations of DNNs provides a new perspective for explanations. The Shapley value [31] estimates the attribution distribution over all players in a game and has been applied to quantitative evaluation of the representations of DNNs [1, 9, 20]. Based on Shapley values, some studies have investigated the interaction between input variables of a DNN [11, 21, 38, 53]. In comparison, our study aims to illustrate distinctive properties of 3D point cloud processing.

## 3  Analyzing feature representations of DNNs for 3D point cloud processing

We propose six metrics to measure regional sensitivities to rotation, translation, scale, and three types of local 3D structures, respectively, so as to exhibit the representation property of each specific point cloud region. Beyond these metrics, we can further analyze the representation quality from the following two perspectives, *i.e.* the spatial smoothness of knowledge representations, and the contextual complexity of 3D structures encoded by a DNN.

**Preliminaries: quantifying the regional attribution using Shapley values.** The Shapley value was originally introduced in game theory [31]. Considering a game with multiple players, each player aims to pursue a high award for victory. The Shapley value is widely considered as a unique unbiased approach that allocates the total reward to each player fairly, **which satisfies axioms of *linearity*, *nullity*, *symmetry*, and *efficiency*** [44] as the theoretical foundation. Please see our supplementary materials for details.

Given a point cloud with $n$ regions[1], $N = \{1, 2, \cdots, n\}$, the Shapley value can be used to measure the attribution of each input region. We can consider the prediction process of a DNN as a game $v$, and each region in a point cloud as a player $i$. Let $2^N \stackrel{\text{def}}{=} \{S \mid S \subseteq N\}$ denote all the possible subsets of $N$. Let $x_S$ denote the point cloud only containing regions in $S$, in which regions in $N \backslash S$ are removed. For a DNN learned for multi-category classification, we use $v(S)$ to denote the network output given the input $x_S$. $v(S)$ is calculated as $\log \frac{p}{1-p}$, where $p = p(y = y^{\text{truth}} \mid x_S)$ denotes the probability of the ground-truth category given $x_S$. Note that the number of input points for each DNN is a fixed value. Therefore, in order to make the DNN successfully handle the point cloud $x_S$ without regions in $N \backslash S$, we reset coordinates of points in regions of $N \backslash S$ to the center of the entire point cloud to remove the information of these points, instead of simply deleting these points [55]. Note that the reason for not simply deleting these points is that the number of input points must be greater than the number of points required for specific layerwise operations of DNNs used in this paper. Specifically, the sampling operation at the first layer of PointNet++ and PointConv requires the minimum number of input points to be greater than 512 points. The $k$-NN-based grouping [43] operation of DGCNN and GCNN requires the number of input points to be greater than 20 points. However, the computation of Shapley values requires to make inference on a point cloud fragment with a single region $i$ (*i.e.,* $x_{\{i\}}$), which may have less than 20 points. Therefore, for such DNNs, we can not simply delete 3D points from the point cloud, in order to allow DNNs to work normally. To this end, for a fair comparison, we choose the alternative way, *i.e.,* placing all points, which are supposed to be masked, to the center of the point cloud. In this way, the numerical attribution of the $i$-th region to the overall prediction score is estimated by the Shapley value $\phi(i)$, as follows.

$$\phi(i) = \sum\nolimits_{S \subseteq N \backslash \{i\}} \frac{|S|!(n - |S| - 1)!}{n!}(v(S \cup \{i\}) - v(S)), \tag{1}$$

The computation of Equation (1) is NP-hard. Therefore, we use the sampling-based method in [4] to approximate $\phi(i)$.

## 3.1 Quantifying the representation sensitivity of a DNN

To analyze the quality of knowledge representations of a DNN, we define different types of sensitivities using the above regional attribution $\phi(i)$. Specifically, we measure six types of sensitivities, including the rotation sensitivity, translation sensitivity, scale sensitivity, and three types of sensitivity to local 3D structures (edge-like structures, surface-like structures, and mass-like structures). Given an input point cloud $x$, the Shapley value $\phi(i)$ measures the attribution of region $i$ to the network output. The rotation/translation/scale/local-structure sensitivity of this region is quantified as the range of changes of this region's attribution $\phi(i)$ among all potential transformations $\{T\}$ of the rotation/translation/scale/local 3D structure, as follows.

$$\forall\, i \in N = \{1, 2, \cdots, n\}, \quad a_i(x) = \frac{1}{Z}(\max_T \phi_{x' = T(x)}(i) - \min_T \phi_{x' = T(x)}(i)), \tag{2}$$

where $Z = \mathbb{E}_T[\sum_{i \in N} |\phi_{x' = T(x)}(i)|]$ is computed for normalization. Thus, the average sensitivity to all potential transformations $\{T\}$ among all input point clouds $x \in X$ is formulated as follows.

$$sensitivity = \mathbb{E}_{x \in X}\big[\mathbb{E}_{i \in N}[a_i(x)]\big] \tag{3}$$

In implementation, six types of sensitivities are computed as follows.

**Rotation sensitivity** quantifies the vulnerability of the inference caused by the rotation of 3D point clouds. Given an input point cloud $x$, we enumerate all rotations $\boldsymbol{\theta} = [\theta_1, \theta_2, \theta_3]^\top$ from the range of $[-\frac{\pi}{4}, \frac{\pi}{4}]$, *i.e.* given $\boldsymbol{\theta}$, we sequentially rotate the point cloud around the three axes of the 3D coordinates system, thereby obtaining a set of new point clouds $\{x' = T_{\text{rotation}}(x|\boldsymbol{\theta})\}$.

**Translation sensitivity** quantifies the vulnerability of the inference caused by the translation of 3D point clouds. Given an input point cloud $x$, we enumerate all translations $\Delta x = [\Delta x_1, \Delta x_2, \Delta x_3]^\top$ from the range of $[-0.5, 0.5]$ along the three axes of the 3D coordinates system, thereby obtaining a set of new point clouds $\{x' = T_{\text{translation}}(x) = x + \Delta x\}$.

---

[1]There are many ways to segment a 3D point cloud to regions. In our experiments, we first used the farthest point sampling [26] to select $n$ points from each point cloud as centers of $n$ regions. Then, we partitioned the point cloud to $n$ regions by assigning each remaining point to the nearest center point among the selected $n$ center points.

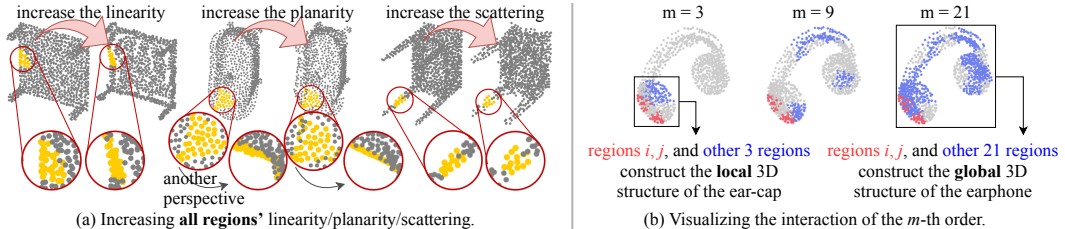

**(a)** Increasing **all regions'** linearity/planarity/scattering.

**(b)** Visualizing the interaction of the *m*-th order.

Figure 2: (a) Visualization of increasing the linearity/planarity/scattering of all regions. (b) Visualization of multi-order interactions.

**Scale sensitivity** quantifies the vulnerability of the inference caused by the scale of 3D point clouds. Given an input point cloud $x$, we enumerate all scales $\alpha$ from the range of $[0.5, 2]$, thereby obtaining a set of new point clouds $\{x' = T_{\text{scale}}(x) = \alpha x\}$.

Then, we focus on **other three sensitivity metrics** for three types of local 3D structures, *i.e.* **sensitivity to linearity (edge-like structures)**, **sensitivity to planarity (surface-like structures)**, and **sensitivity to scattering (mass-like structures)**. According to [12, 6], the significance of a point cloud to be edge-like structures/surface-like structures/mass-like structures is defined as follows.

$$linearity = \frac{\lambda_1 - \lambda_2}{\lambda_1}; \quad planarity = \frac{\lambda_2 - \lambda_3}{\lambda_1}; \quad scattering = \frac{\lambda_3}{\lambda_1}; \tag{4}$$

where $\lambda_1 \geq \lambda_2 \geq \lambda_3$ denote three eigenvalues of the covariance matrix of a region's points.

Without loss of generality, we take the linearity for example to introduce the way to enumerate a region's all values of *linearity*. Given a region, we use the gradient ascent/descent method to modify the 3D coordinates $\mathbf{p} \in \mathbb{R}^3$ of each point in this region, so as to increase/decrease the *linearity*, *i.e.* $\mathbf{p}^{\text{new}} = \mathbf{p} + \eta \frac{\partial linearity}{\partial \mathbf{p}}$. As Fig. 2 (a) shows, we enumerate all regions' *linearity* at the same time. The enumeration is implemented under the following two constraints. (1) Each value of $\lambda_1$ needs to be within the range of $\lambda_1 \pm \gamma$, so as the $\lambda_2$ and $\lambda_3$. (2) $\|\mathbf{p}^{\text{final}} - \mathbf{p}^{\text{ori}}\| \leq d$, where $\|\cdot\|$ denotes the L2-norm. In this way, we obtain a set of point clouds with different degrees of local edge-like structures. In real implementation, we set $\eta = 0.001$, $\gamma = 0.003$, and $d = 0.03$.

## 3.2 Quantifying the spatial smoothness of knowledge representations

Besides the analysis of representation sensitivity, we also evaluate the spatial smoothness of feature representations encoded by a DNN. Because most benchmark 3D datasets [46, 49] only contain objects with simple 3D structures (see Fig. 1 (c)), except for special regions (*e.g.* edges), most adjacent regions of such simple objects usually have similar local 3D structures (*e.g.* surfaces). Therefore, most adjacent regions are supposed to have similar regional attributions, *i.e.* high smoothness. In this way, high spatial smoothness indicates reliable feature representations. We quantify the smoothness of the regional attribution between neighboring regions, as follows.

$$non\text{-}smoothness = \mathbb{E}_{x \in X} \mathbb{E}_T \mathbb{E}_i \mathbb{E}_{j \in \mathcal{N}(i)} \left[ \frac{|\phi_{x'}(i) - \phi_{x'}(j)|}{Z_{\text{smooth}}} \Big|_{x'=T(x)} \right] \tag{5}$$

where $\mathcal{N}(i)$ denotes a set of nearest point cloud regions (determined by the ball-query search [28]) of region $i$; $Z_{\text{smooth}} = \mathbb{E}_T[|v_{x'}(N) - v_{x'}(\emptyset)|_{x'=T(x)}]$ is computed for normalization; $v_{x'}(N)$ denotes the network output given the entire point cloud $x'$, and $v_{x'}(\emptyset)$ denotes the network output when we mask all regions in $x'$ by setting all points to the center coordinates of the point cloud.

## 3.3 Quantifying the interaction complexity of a DNN

In addition, we also quantify the representation complexity of a DNN, *i.e.* the complexity of 3D structures encoded by a DNN. To this end, the 3D structures encoded by a DNN is represented using the interaction between different 3D point cloud regions. Here, input regions of a DNN do not work individually, but collaborate with each other to construct a specific 3D structure for inference. Zhang *et al.* [53] defined the multi-order interactions between two input variables. Given two input regions $i$ and $j$, the interaction of the $m$-th order measures the additional attribution brought by collaborations

Table 1: Classification accuracy of different DNNs.

| Model | PointNet | PointNet++ | PointConv | DGCNN | GCNN | adv-GCNN[4] | RotationNet [14] | 3DmFV-Net [3] | KCNet [34] | GIFT [2] |
|---|---|---|---|---|---|---|---|---|---|---|
| ModelNet10 | 93.5 | 94.7 | 94.6 | 94.4 | 95.1 | 91.0 | 98.46 | 95.2 | 94.4 | 92.35 |
| ShapeNet part | 99.2 | 99.7 | 99.5 | 100.0 | 100.0 | 97.9 | – | – | – | – |

between $i$ and $j$ under the context of $m$ regions.

$$I^{(m)}(i,j) = \mathbb{E}_{S \subseteq N \setminus \{i,j\}, |S|=m} \big[ v(S \cup \{i,j\}) - v(S \cup \{i\}) - v(S \cup \{j\}) + v(S) \big], \qquad (6)$$

$I^{(m)}(i,j) > 0$ indicates that the presence of region $j$ increases the attribution of region $i$, $\phi(i)$. $I^{(m)}(i,j) < 0$ indicates that the presence of region $j$ decreases the value of $\phi(i)$. $I^{(m)}(i,j) \approx 0$ indicates that region $j$ and region $i$ are almost independent of each other. Please see our supplemental materials for details about the meaning of multi-order interaction and the *linearity*, *nullity*, *commutativity*, *symmetry*, and *efficiency* axioms of the multi-order interaction.

Here, we could consider the order $m$ as the number of contextual regions involved in the computation of interactions between region $i$ and region $j$. For example, as Fig. 2 (b) shows, regions $i$ and $j$ (in red), and other $m$ regions (in blue) work together to construct a 3D structure to classify the earphone.

High-order interactions measure the effects of global collaborations among massive regions, *i.e.* representing complex and large-scale 3D structures. Low-order interactions measure the effects of collaborations between a few regions, *i.e.* usually representing simple and small-scale 3D structures. Then, we use the following metric to quantify the average strength of the $m$-th order interactions as the significance of the $m$-order complex 3D structures.

$$I^{(m)} = \mathbb{E}_{x \in X} \Big[ \big| \mathbb{E}_{i,j}[I_x^{(m)}(i,j)] \big| \Big]. \qquad (7)$$

If the $I^{(m)}$ of a low order is significantly larger than that of a high order, then the representation complexity of the DNN is limited to representing simple and local 3D structures.

## 4 Comparative studies

In this section, we conducted comparative studies to analyze properties of different point cloud regions of different DNNs. Ideally, a well-trained DNN for 3D point cloud processing was supposed to be robust to the rotation and translation, and the DNN was supposed to mainly use the scale and local 3D structures for inference. Besides, considering the 3D structures of objects in benchmark 3D datasets [46, 49] were usually simple, most adjacent regions in an object had continuous and similar 3D structures. Therefore, a well-trained DNN was supposed to have similar regional attributions among these neighboring regions. We also analyzed how complex is the 3D structure that can be encoded by a classic DNN.

We used our method to analyze five classic DNNs for 3D point cloud processing, including the PointNet [26], the PointNet++ [28], the DGCNN [43], the non-dynamic version of DGCNN (*i.e.* GCNN), and the PointConv [45]. All DNNs were learned based on the ModelNet10 dataset [46] and the ShapeNet part[2] dataset [49]. We followed [26] to only use 1024 points of each point cloud to train all DNNs. Each point cloud was partitioned to $n = 32$ regions[1] for the computation of all metrics.

Note that all DNNs used in our paper were well-trained. The testing accuracy was shown in Table 1.We also reported the testing accuracy of other DNNs trained on the ModelNet10 dataset, including RotationNet [14], 3DmFV-Net [3], KCNet [34], and GIFT [2]. These four DNNs obtained the 1st, 4th, 7th, and 10th testing accuracy according the ModelNet10 Benchmark Leaderboard[3].

In particular, the previous study [54] has discovered that people could use rotations to attack DNNs for 3D point cloud processing. Therefore, we used the adversarial training to learn a GCNN[4] *w.r.t.* the attacks based on rotations and translations of point clouds, so as to improve the GCNN's robustness to the rotation and translation. We extended the method in [16] to generate such adversarial examples to attack the GCNN. Please see our supplemental materials for details. The objective function

---

[2]We only used part of the ShapeNet part dataset due to the time limitation. Please see our supplemental materials for details.

[3]Data from https://modelnet.cs.princeton.edu/.

[4]In tables, adv-GCNN denoted the adversarially trained GCNN.

Table 2: Average sensitivities over all regions among all samples.

| Dataset | Model | rotation sensitivity | translation sensitivity | scale sensitivity | sensitivity to edges | sensitivity to surfaces | sensitivity to masses |
|---|---|---|---|---|---|---|---|
| ModelNet10 | PointNet | 0.159±0.070 | 0.110±0.053 | 0.024±0.017 | 0.007±0.007 | 0.010±0.009 | 0.009±0.009 |
| | PointNet++ | 0.171±0.064 | 0.004±0.004 | 0.054±0.027 | 0.018±0.011 | 0.026±0.016 | 0.029±0.019 |
| | PointConv | 0.145±0.060 | 2.3e-4±1.9e-4 | 0.027±0.019 | 0.010±0.007 | 0.015±0.011 | 0.017±0.013 |
| | DGCNN | 0.174±0.075 | 0.048±0.024 | 0.020±0.014 | 0.016±0.009 | 0.022±0.014 | 0.023±0.015 |
| | GCNN | 0.174±0.067 | 0.050±0.026 | 0.020±0.014 | 0.017±0.010 | 0.022±0.014 | 0.023±0.015 |
| | adv-GCNN[4] | 0.034±0.012 | 0.007±0.004 | 0.020±0.014 | 0.022±0.014 | 0.027±0.014 | 0.029±0.018 |
| ShapeNet part | PointNet | 0.107±0.065 | 0.071±0.032 | 0.023±0.020 | 0.005±0.005 | 0.004±0.004 | 0.005±0.005 |
| | PointNet++ | 0.142±0.057 | 0.001±0.000 | 0.044±0.025 | 0.014±0.009 | 0.014±0.009 | 0.016±0.011 |
| | PointConv | 0.168±0.073 | 1.4e-5±2.5e-5 | 0.053±0.042 | 0.017±0.013 | 0.016±0.011 | 0.019±0.015 |
| | DGCNN | 0.141±0.069 | 0.067±0.033 | 0.020±0.015 | 0.014±0.011 | 0.013±0.011 | 0.016±0.013 |
| | GCNN | 0.141±0.065 | 0.072±0.038 | 0.021±0.015 | 0.014±0.011 | 0.013±0.010 | 0.016±0.015 |
| | adv-GCNN[4] | 0.028±0.012 | 0.009±0.008 | 0.025±0.020 | 0.028±0.022 | 0.024±0.015 | 0.028±0.019 |

Table 3: Pearson correlation coefficients between regional attributions and sensitivities.

| Models | ModelNet10 dataset | | | ShapeNet part dataset | | |
|---|---|---|---|---|---|---|
| | rotation sensitivity | translation sensitivity | scale sensitivity | rotation sensitivity | translation sensitivity | scale sensitivity |
| PointNet | 0.648±0.266 | 0.637±0.165 | 0.473±0.194 | 0.528±0.278 | 0.549±0.204 | 0.538±0.275 |
| PointNet++ | 0.811±0.123 | 0.415±0.189 | 0.592±0.142 | 0.629±0.154 | 0.266±0.269 | 0.543±0.171 |
| PointConv | 0.601±0.234 | 0.009±0.179 | 0.473±0.174 | 0.739±0.166 | -0.006±0.170 | 0.617±0.168 |
| DGCNN | 0.788±0.111 | 0.622±0.164 | 0.494±0.224 | 0.725±0.176 | 0.649±0.174 | 0.458±0.201 |
| GCNN | 0.832±0.082 | 0.610±0.131 | 0.464±0.231 | 0.696±0.158 | 0.682±0.198 | 0.431±0.199 |
| adv-GCNN[4] | 0.488±0.167 | 0.298±0.234 | 0.414±0.256 | 0.343±0.234 | 0.255±0.223 | 0.476±0.304 |

was $\min_w \mathbb{E}_x \left[ \max_T Loss(x' = T(x), y^{\text{truth}}; w) \right]$, where $w$ denoted the parameter of the GCNN. Comparative studies have revealed the following effects of the adversarial training (using rotations and translations for attack, instead of using perturbations) on knowledge representations. The adversarial training shifted the attention of DNN from orientations and positions to structural information (including global structures in normal samples, see Fig. 5), thereby increasing the sensitivity to 3D structures (see Table 2 and Fig. 3 (a)). The adversarial training also decreased the correlation between the regional sensitivity and the regional attribution (see Table 3), and increased the spatial smoothness of knowledge representations (see Table 4).

**Comparative study 1, explaining the regional sensitivity of DNNs.** Table 2 shows six types of regional sensitivities of six DNNs learned on the ModelNet10 dataset and the ShapeNet part dataset. In terms of translation sensitivity, PointNet was relatively sensitive to the translation, because PointNet extracted features from global coordinates of points, thereby being sensitive to the translation. In contrast, PointNet++ and PointConv were robust to translation, because these two DNNs encoded relative coordinates. Particularly, PointConv did not use global coordinates during the inference process, thereby yielding the lowest translation sensitivity (see the point cloud with the darkest blue in the second row of Fig. 3 (a)). In terms of scale sensitivity, PointNet++ was relatively sensitive to the scale, because PointNet++ encoded features of neighborhood with fixed scales. Besides, because adversarial training forced the GCNN to remove attention from rotation-sensitive features and translation-sensitive features, the adversarially trained GCNN paid more attention to structural information. Therefore, compared with the original GCNN, the adversarially trained GCNN had higher sensitivity to local 3D structures. Furthermore, we also obtained the following conclusions.

• *Rotation robustness was the Achilles' heel of classic DNNs for 3D point cloud processing.* As Table 2 and Fig. 3 (a) show, all DNNs were sensitive to rotations except for the adversarially trained GCNN.

• *PointNet failed to encode local 3D structures.* In terms of sensitivity to local 3D structures, PointNet was the least sensitive DNN (see Table 2). It was because convolution operations of the PointNet encoded the information of each point independently, *i.e.* the PointNet did not encode the information of neighboring points/regions. This conclusion is also verified by the phenomenon that PointNet had darker blue point clouds than other DNNs (see the last three rows of Fig. 3 (a)).

• *Most DNNs usually failed to extract rotation-robust features from 3D points at edges and corners.* Given a point cloud $x$, we selected the most rotation-sensitive region $i^*$ (shown as black boxes in Fig. 3 (b)). We rotated the point cloud $x$ to the orientations that maximized and minimized the

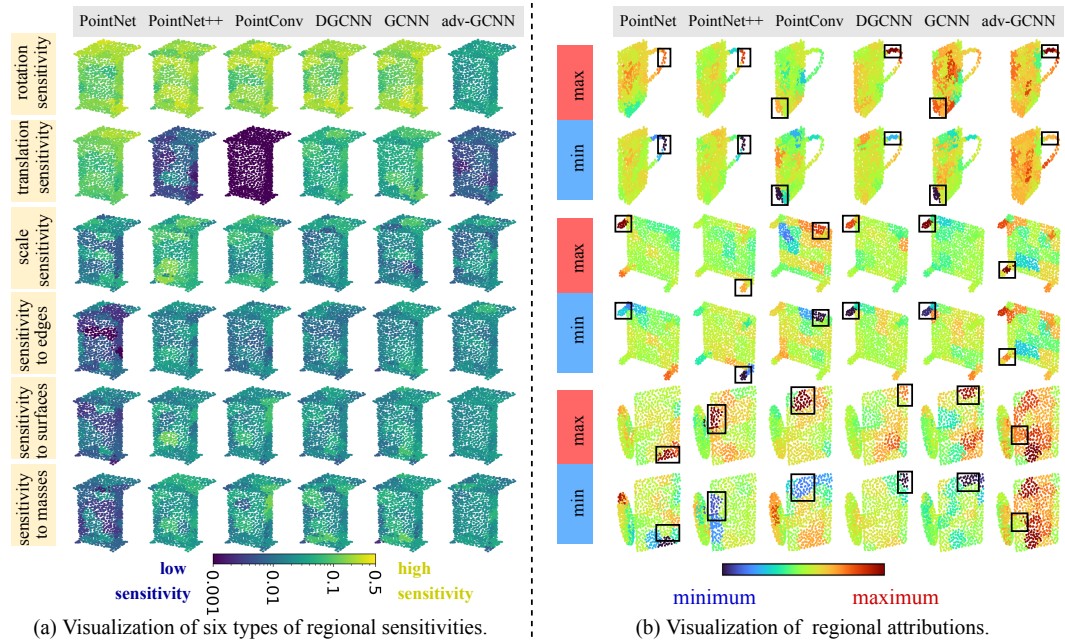

(a) Visualization of six types of regional sensitivities.

(b) Visualization of regional attributions.

Figure 3: Visualization of regional sensitivities and regional attributions. (a) Visualization of regional sensitivities. The regional sensitivities of all point clouds are normalized to the same colorbar, which is shown in a log-scale. (b) Visualization of regional attributions. For each point cloud, we selected the most rotation-sensitive region $i^*$ (shown as black boxes) and visualized the pair of point clouds with specific orientations corresponding to the maximum and the minimum regional attributions of the region $i^*$. More visualization results are shown in our supplemental materials.

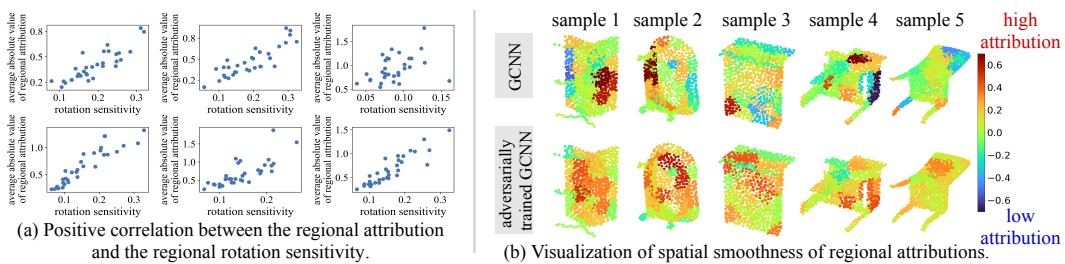

(a) Positive correlation between the regional attribution and the regional rotation sensitivity.

(b) Visualization of spatial smoothness of regional attributions.

Figure 4: (a) Positive correlation between the regional rotation sensitivity and the regional attribution of the PointNet. (b) Visualization of spatial smoothness of regional attributions. Experimental results show that the adversarial training increased the smoothness of neighboring regions' attributions.

attribution of region $i^*$, *i.e.* $\boldsymbol{\theta}_1 = \arg\max_{\boldsymbol{\theta}} \phi_{x'}(i^*)$ and $\boldsymbol{\theta}_2 = \arg\min_{\boldsymbol{\theta}} \phi_{x'}(i^*)$. Fig. 3 (b) visualizes regional attributions of point clouds rotated by $\boldsymbol{\theta}_1$ and $\boldsymbol{\theta}_2$. We found that rotation-sensitive regions were usually distributed on the edges and corners, which verified our conclusion.

● *Most DNNs usually could not ignore features of rotation-sensitive points at edges and corners.* Table 3 shows that the Pearson correlation coefficient [25] between each region's average strength of attribution over different rotations (*i.e.* $\mathbb{E}_{\boldsymbol{\theta}}[|\phi_{x'}(i)|]$, s.t. $x' = T_{\text{rotation}}(x|\boldsymbol{\theta})$) and this region's rotation sensitivity $a_i(x)$ is much larger than Pearson correlation coefficients between the attribution and other sensitivities. This means that rotation-sensitive regions (which were usually distributed on the edges and corners) usually had large regional attributions, which verified our conclusion. Besides, Table 3 also shows that the adversarial training reduced the correlation between the regional sensitivity and the regional attribution. Fig. 4 (a) visualizes the positive correlation between the regional rotation sensitivity and the regional attribution of the PointNet.

**Comparative study 2, explaining the spatial smoothness of knowledge representations.** Table 4 shows the non-smoothness of knowledge representations encoded by different DNNs. Note that the adversarially trained GCNN was significantly biased to the *knife* category. In other words, the

Table 4: The non-smoothness of attributions between adjacent regions.

| Models | ModelNet10 dataset | | ShapeNet part dataset | | ShapeNet part dataset (removing the biased category) | |
|---|---|---|---|---|---|---|
| | rotation | translation | rotation | translation | rotation | translation |
| PointNet | 0.071±0.039 | 0.029±0.017 | 0.025±0.009 | 0.016±0.005 | 0.025±0.010 | 0.015±0.003 |
| PointNet++ | 0.091±0.041 | 0.041±0.022 | 0.036±0.011 | 0.022±0.016 | 0.034±0.010 | 0.017±0.003 |
| PointConv | 0.047±0.014 | 0.056±0.108 | 0.080±0.019 | 0.040±0.017 | 0.081±0.020 | 0.039±0.018 |
| DGCNN | 0.071±0.024 | 0.031±0.010 | 0.047±0.019 | 0.026±0.017 | 0.044±0.017 | 0.021±0.005 |
| GCNN | 0.083±0.026 | 0.034±0.012 | 0.050±0.019 | 0.027±0.010 | 0.049±0.020 | 0.025±0.008 |
| adv-GCNN[4] | 0.029±0.012 | 0.030±0.013 | 0.054±0.110 | 0.056±0.114 | 0.022±0.008 | 0.023±0.008 |

adversarially trained GCNN classified the empty input as the *knife* category with a high confidence (please see our supplementary materials for details). Therefore, to enable fair comparisons, we also reported the spatial non-smoothness of the adversarially trained GCNN on all other categories except the *knife*. We discovered that *without considering the biased knife category, adversarial training increased the spatial smoothness of knowledge representations.* Fig 4 (b) also verified this conclusion.

**Comparative study 3, explaining the interaction complexity of DNNs.** Fig. 5 shows multi-order interactions of different DNNs. From this figure, we discovered the following new insights.

• *Most DNNs failed to encode high-order interactions (*i.e. *global and large-scale 3D structures).* As Fig. 5 (a) shows, no matter given normal samples or adversarial samples[5], classic DNNs encoded extremely low-order interactions. This indicated that most DNNs did not extract complex and large-scale 3D structures from normal samples.

• *The adversarial training (using rotations and translations for attack, instead of using perturbations) increased the effects of extremely high-order interactions.* As Fig. 5 (a) shows, the adversarially trained GCNN encoded extremely low-order interactions and extremely high-order interactions (*i.e.* the global structures of objects) from normal samples. Because the GCNN was forced to sophisticatedly select relatively complex rotation-robust features among all potential features.

• *Rotation sensitivity of regions with out-of-distribution high-order interactions (i.e. abnormal complex and large-scale 3D structures) were larger than the average sensitivity over all regions.* In fact, there are two types of high-order interactions. Unlike the above two insights describing whether or not a DNN encoded high-order interactions **on normal samples**, this insight mainly focused on abnormal out-of-distribution high-order interactions in a DNN. When a point cloud was attacked by adversarial rotations[5], the adversarial rotation[5] would generate out-of-distribution high-order interactions (*i.e.* abnormal complex and large-scale 3D structures) to attack the DNN. We measured interactions between the most rotation-sensitive region $i^*$ and its neighbors $j \in \mathcal{N}(i^*)$ among all input point clouds, *i.e.* $\mathbb{E}_{x \in X}\left[|\mathbb{E}_{j \in \mathcal{N}(i^*)}[I_x^{(m)}(i^*, j)]|\right]$. As Fig. 5 (b) shows, rotation-sensitive regions usually encoded relatively more high-order interactions than the overall interaction distribtuion over all regions. This indicates that compared to most other regions, rotation-sensitive regions paid more attention to high-order interactions, although low-order interactions of rotation-sensitive regions increased too.

**Broad applicability of the proposed metrics.** We also conducted experiments on an AutoEncoder for 3D point cloud reconstruction task and rotation-invariant DNNs for 3D point cloud classification, in order to further demonstrate the broad applicability of the proposed metrics.

For the AutoEncoder for 3D point cloud reconstruction task, we used all layers before the fully-connected layers in PointNet [26] as the encoder and used the decoder in the FoldingNet [48] as the decoder. We took $v(S) = \frac{(z(N)-z(\emptyset))^\top (z(S)-z(\emptyset))}{||z(N)-z(\emptyset)||_2}$ as the reward score in Equation (1), where $z(S)$ denotes the output vector of the encoder given the input point cloud consisting of regions in $S$. This reward score measures the utility of the encoder output $z(S)$ given regions in $S$ along the direction of $z(N) - z(\emptyset)$. The AutoEncoder was learned based on the ModelNet10 dataset. We measured the rotation sensitivity (0.099±0.043), the translation sensitivity (0.152±0.086), and the scale sensitivity (0.045±0.042) of the AutoEncoder. Compared with results in Table 2, we found that (1) the AutoEncoder for reconstruction was also sensitive to rotation; (2) the AutoEncoder for 3D

---

[5]Here, we used rotations for attack, instead of using perturbations.

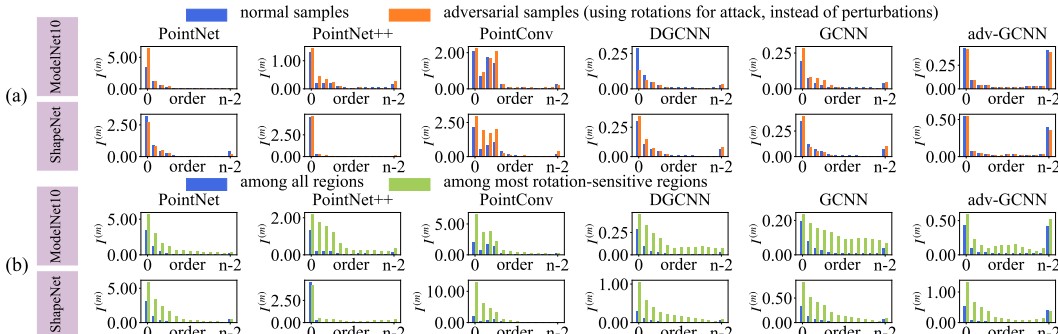

Figure 5: Multi-order interactions of different DNNs. (a) Comparison of multi-order interactions between normal samples and adversarial samples[5]. (b) Comparison of multi-order interactions between all regions and rotation-sensitive regions.

Table 5: Sensitivities of DNNs with or without different types of augmentation. For DNNs with or without translation/scale/rotation augmentation, we reported the translation/scale/rotation sensitivity.

| Network architecture | translation aug | | scale aug | | rotation aug around the y-axis | | rotation aug around a random axis | |
|---|---|---|---|---|---|---|---|---|
| | w/ | w/o | w/ | w/o | w/ | w/o | w/ | w/o |
| PointNet | 0.111±0.053 | 0.160±0.067 | 0.025±0.017 | 0.063±0.047 | 0.108±0.047 | 0.155±0.068 | 0.057±0.030 | 0.155±0.068 |
| DGCNN | 0.048±0.024 | 0.098±0.054 | 0.020±0.015 | 0.028±0.020 | 0.158±0.063 | 0.173±0.072 | 0.052±0.021 | 0.173±0.072 |

point cloud reconstruction was more sensitive to translation than the DNN for classification; (3) the AutoEncoder for reconstruction was sensitive to the scale change.

Besides, we followed [32] to revise PointNet++ and DGCNN to be rotation-invariant DNNs. We measured the rotation sensitivity of the rotation-invariant PointNet++ (0.002±0.001) and DGCNN (0.010±0.004). Compared with results in Table 2, we found that rotation-invariant DNNs were much more robust to rotations than traditional DNNs (even the adversarially-trained DNN).

**Effects of data augmentation on sensitivities.** We conducted comparative studies to explore the effects of data augmentation on sensitivities, including translation augmentation, scale augmentation, rotation augmentation around the y-axis, and rotation augmentation around a random axis. We conducted experiments on PointNet and DGCNN. To explore the effects of each of the above augmentation on sensitivities, we learned two versions of each network architecture, *i.e.* one network with the specific augmentation and the other network without the specific augmentation. Please see our supplementary materials for more details about different versions of each network architecture. All DNNs were learned based on the ModelNet10 dataset. Table 5 shows that rotation/translation/scale augmentation decreased the rotation/translation/scale sensitivity of a DNN.

Besides, we compared the effects of rotation augmentation around a random axis and the effects of adversarial training on the rotation sensitivity. We conducted experiments on DGCNN and GCNN. We trained two versions of DGCNN and GCNN, one with rotation augmentation around a random axis (DGCNN, 0.052 ±0.021; GCNN, 0.048±0.024) and one with the adversarial training using rotations for attack (DGCNN, 0.036±0.015; GCNN, 0.028±0.016). We found that compared with the rotation augmentation, the adversarial training *w.r.t.* rotation-based attacks had a greater impact on the rotation sensitivity.

## 5   Conclusion

In this paper, we have measured six types of regional sensitivities, the spatial smoothness, and the contextual complexity of feature representations encoded by DNNs. Comparative studies have discovered several new insights into classic DNNs for 3D point cloud processing. We have found that most DNNs were extremely sensitive to the rotation. Rotation-sensitive regions were usually distributed on the edges and corners of objects, had large attributions to network output, and usually paid more attention to abnormal large-scale structures. In addition, most DNNs failed to extract complex and large-scale 3D structures from normal samples, while the adversarial training could encourage the DNN to extract global structures from normal samples.

## Acknowledgments and Disclosure of Funding

This work is partially supported by the National Nature Science Foundation of China (No. 61906120, U19B2043), Shanghai Natural Science Fundation (21JC1403800,21ZR1434600), Shanghai Municipal Science and Technology Major Project (2021SHZDZX0102). This work is also partially supported by Huawei Technologies Inc.

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
