# Interpreting Representation Quality of DNNs for 3D Point Cloud Processing: Supplementary Materials

**Wen Shen**[b*]     **Qihan Ren**[a]     **Dongrui Liu**[a]     **Quanshi Zhang**[a†]

[a]Shanghai Jiao Tong University     [b]Tongji University

## A   Shapley values

This section provides more details about Shapley values in Section 3 of the paper. The Shapley value $\phi(i) = \sum_{S \subseteq N \setminus \{i\}} \frac{|S|!(n-|S|-1)!}{n!}(v(S \cup \{i\}) - v(S))$ satisfies axioms of *linearity*, *nullity*, *symmetry*, and *efficiency* [5], as follows.

• *Linearity*: If two independent games $v$ and $w$ can be merged into one game $u(S) = v(S) + w(S)$, then the Shapley value of the player $i$ in game $v$ and game $w$ also can be merged, *i.e.* $\phi_u(i) = \phi_v(i) + \phi_w(i)$.

• *Nullity*: A dummy player $i$ satisfies $\forall S \subseteq N \setminus \{i\}, v(S \cup \{i\}) = v(S) + v(\{i\})$, which indicates that the player $i$ has no interaction with other players, *i.e.* $\phi(i) = v(\{i\})$.

• *Symmetry*: Given two players $i, j$, if $\forall S \subseteq N \setminus \{i, j\}, v(S \cup \{i\}) = v(S \cup \{j\})$, then $\phi(i) = \phi(j)$.

• *Efficiency*: The overall reward can be allocated to all players in the game, *i.e.* $\sum_{i \in N} \phi(i) = v(N) - v(\emptyset)$.

## B   Multi-order interactions

This section provides more details about multi-order interactions [8] in Section 3.3 of the paper. Given two input variables $i$ and $j$ (in this paper, input variables indicate point cloud regions), the interaction of the $m$-th order measures the additional attribution brought by collaborations between regions $i$ and $j$ under the context of $m$ regions:

$$I^{(m)}(i,j) = \mathbb{E}_{S \subseteq N \setminus \{i,j\}, |S|=m}\big[v(S \cup \{i,j\}) - v(S \cup \{i\}) - v(S \cup \{j\}) + v(S)\big]. \quad (1)$$

$I^{(m)}(i,j) > 0$ indicates that the presence of region $j$ increases the attribution of region $i$, $\phi(i)$, under the context of other $m$ regions. $I^{(m)}(i,j) < 0$ indicates that the presence of region $j$ decreases the value of $\phi(i)$. $I^{(m)}(i,j) \approx 0$ indicates that region $j$ and region $i$ are almost independent. When $m$ is small, $I^{(m)}(i,j)$ reflects the interaction between $i$ and $j$ *w.r.t.* simple contextual collaborations with a few regions. When $m$ is large, $I^{(m)}(i,j)$ corresponds to the interaction *w.r.t.* complex contextual collaborations with massive regions.

The multi-order interaction satisfies axioms of *linearity*, *nullity*, *commutativity*, *symmetry*, and *efficiency* [8], as follows.

---

[*]This work was done when Wen Shen was an intern at Shanghai Jiao Tong University.

[†]Quanshi Zhang is the corresponding author. This study was done under the supervision of Dr. Quanshi Zhang. He is with the John Hopcroft Center and the MoE Key Lab of Artificial Intelligence, AI Institute, at the Shanghai Jiao Tong University, China.

35th Conference on Neural Information Processing Systems (NeurIPS 2021).

- *Linearity*: If two independent games $v$ and $w$ can be merged into one game $u(S) = v(S) + w(S)$, then the interaction of the player $i$ in game $v$ and game $w$ also can be merged, *i.e.* $I_u^{(m)}(i,j) = I_v^{(m)}(i,j) + I_w^{(m)}(i,j)$.

- *Nullity*: A dummy player $i$ satisfies $\forall S \subseteq N\backslash\{i\}, v(S \cup \{i\}) = v(S) + v(\{i\})$, which indicates that the player $i$ has no interaction with other players, *i.e.* $\forall j, I^{(m)}(i,j) = 0$.

- *Commutativity*: $\forall i, j, I^{(m)}(i,j) = I^{(m)}(j,i)$.

- *Symmetry*: If two players $i, j$ have same collaborations with other players $\forall S \subseteq N\backslash\{i,j\}, v(S \cup \{i\}) = v(S \cup \{j\})$, then $\forall k \in N, I^{(m)}(i,k) = I^{(m)}(j,k)$.

- *Efficiency*: The overall reward can be decomposed into interactions of different orders, *i.e.* $v(N) = v(\emptyset) + \sum_{i\in N}(v(\{i\}) - v(\emptyset)) + \sum_{i\in N}\sum_{j\in N\backslash\{i\}}[\sum_{m=0}^{n-2}\frac{n-1-m}{n(n-1)}I^{(m)}(i,j)]$.

## C  More visualization results

This section provides more visualization results of regional sensitivities (see Fig. 1) and regional attributions (see Fig. 2). These visualization results help us understand the conclusions in "**Comparative study 1, explaining the regional sensitivity of DNNs.**" in Section 4 of the paper.

## D  Technical details of combining [4] and [1]

In this section, we provide technical details of the method in [1], based on which we adversarially trained a GCNN [4] *w.r.t.* the attacks based on rotations and translations of point clouds, so as to improve the GCNN's robustness to rotation and translation. Considering a GCNN with model parameters $w$ and loss function *Loss*, let $x$ denote the input point cloud and $y^{\text{truth}}$ denote the ground truth label. The objective function was

$$\min_w \mathbb{E}_x \left[ \max_T Loss(x' = T(x), y^{\text{truth}}; w) \right], \tag{2}$$

where $T$ denoted the transformation applied on the input point cloud $x$, *e.g.* rotations or translations. We followed [1] to generate adversarial samples. The method in [1] applied the following update rule to generate adversarial samples through multiple iterations.

$$x_{\text{adv}}^{(0)} = x, \quad x_{\text{adv}}^{(t+1)} = Clip_\epsilon \left\{ x_{\text{adv}}^{(t)} + \eta \, \text{sign}(\nabla_x Loss(x_{\text{adv}}^{(t)}, y^{\text{truth}})) \right\}, \tag{3}$$

where $\eta$ was the step size, $Clip_\epsilon$ denoted the clipping operation, so that $x_{\text{adv}}^{(t)}$ will be in $L_\infty$ $\epsilon$-neighborhood of $x$. We followed Equation (3) to generate adversarial rotations and translations, and then based on such adversarial rotations and translations to generate adversarial samples, so as to adversarially train a GCNN.

## E  About the ShapeNet part dataset

This section provides more details about the use of the ShapeNet part dataset in the paper. We trained DNNs based on the ShapeNet part dataset [7], which contained 16,881 point clouds from 16 categories. Due to time limitation, we removed six categories with the largest number of samples, and only used the remaining ten categories to train DNNs and to evaluate the proposed metrics. The remaining ten categories are *Bag*, *Cap*, *Earphone*, *Knife*, *Laptop*, *Motorbike*, *Mug*, *Pistol*, *Rocket*, and *Skateboard*.

## F  Bias of the *Knife* category

In Comparative study 2 in Section 4 in the paper, we claimed that the adversarially trained GCNN based on the ShapeNet part dataset was significantly biased to the *knife* category. In this section in the supplementary material, we discuss more about this phenomenon and explore whether this phenomenon (biased knife category) is a specific situation or an accident.

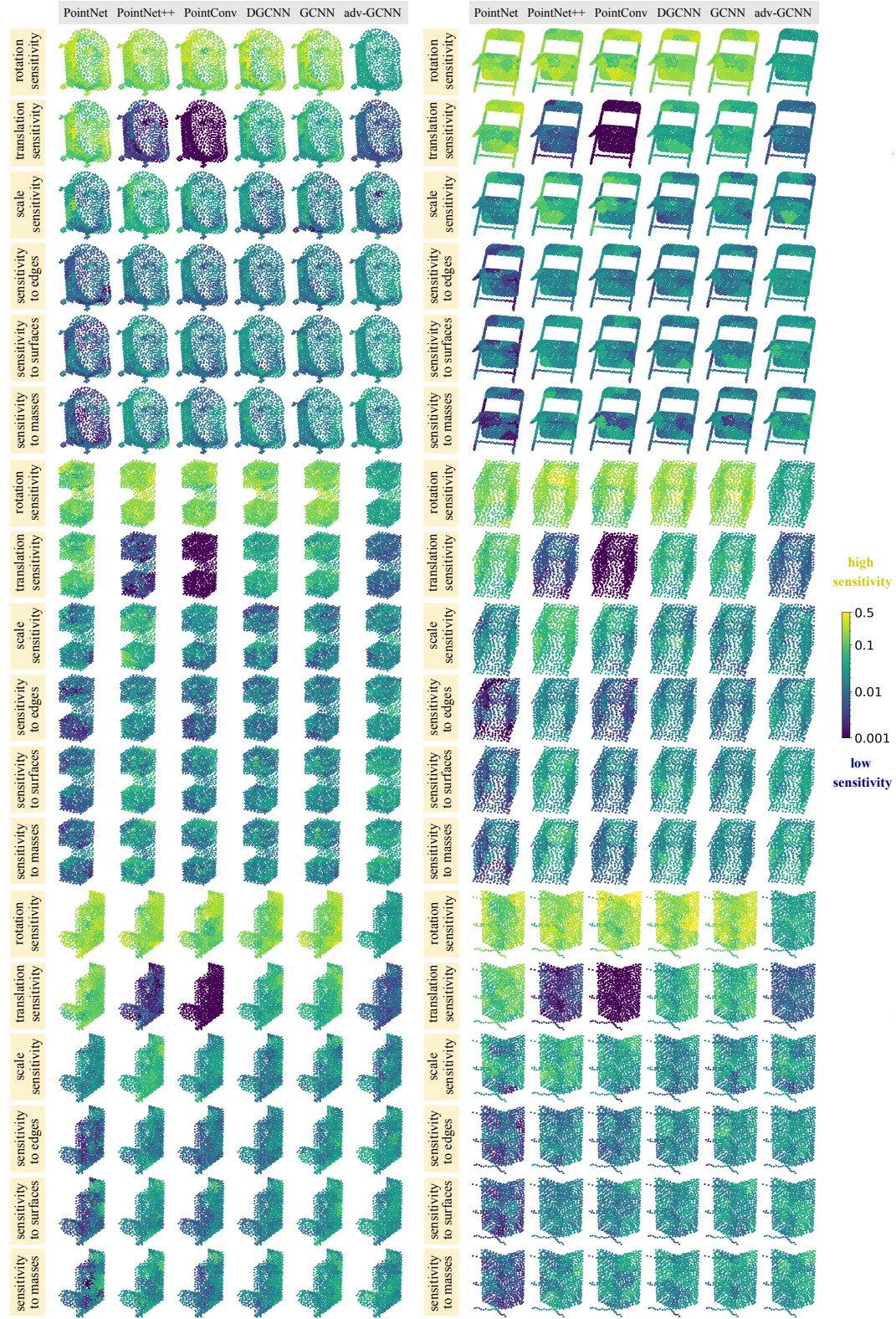

Figure 1: Visualization of regional sensitivities. The regional sensitivities of all point clouds are normalized to the same colorbar, which is shown in a log-scale.

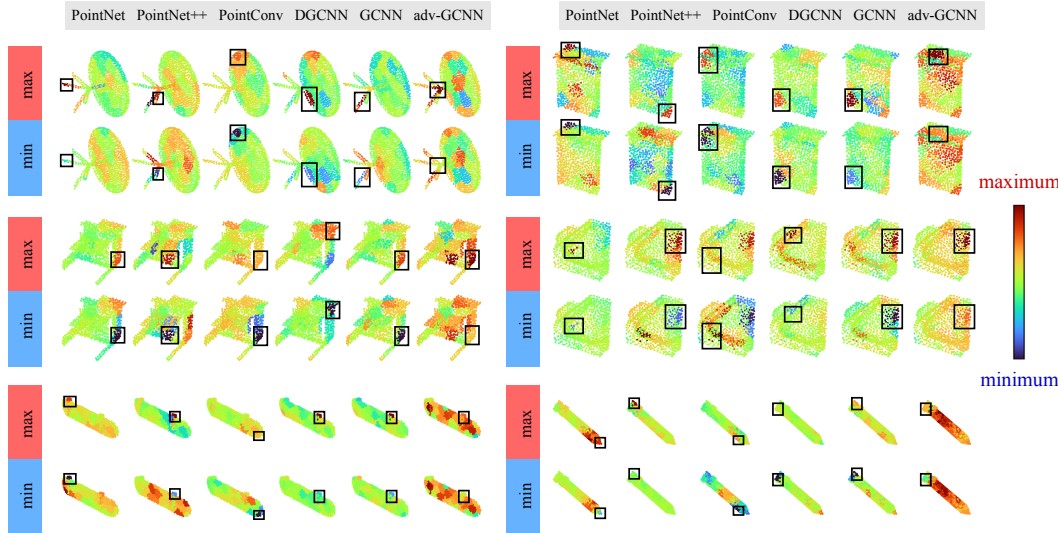

Figure 2: Visualization of regional attributions. For each point cloud, we selected the most rotation-sensitive region $i^*$ (shown as black boxes) and visualized the pair of point clouds with specific orientations corresponding to the maximum and the minimum regional attributions of the region $i^*$.

Table 1: Different versions of a specific network architecture using different data augmentation.

| Model | translation data augmentation | scale data augmentation | rotation data augmentation around the y-axis | rotation data augmentation around a random axis |
|---|---|---|---|---|
| A: baseline settings in [4] | ✓ | ✓ | × | × |
| B | × | ✓ | × | × |
| C | ✓ | × | × | × |
| D | ✓ | ✓ | ✓ | × |
| E | ✓ | ✓ | × | ✓ |

The bias to the *knife* category means that the probability of the *knife* category was almost the same when given the empty input (we reset coordinates of all points in the input to the center of the entire point cloud to obtain the empty input) and an entire point cloud in the *knife* category, which indicated that adversarially trained GCNN did not learn the specific knowledge to classify the *knife* category.

To explore whether the above phenomenon (biased knife category) is a specific situation or an accident, we explored the bias appearing in the adversarially-trained DNNs with different initial parameters. If the bias of different DNNs was always the same (*i.e.,* all DNNs classified an empty input as the knife category), then it meant that the knife was a special category that the empty point cloud was classified as. If the bias of different DNNs was different, then it meant that this phenomenon was an accident.

We trained the adversarially-trained GCNN five times under different initial parameters. The biased category appearing in different DNNs was always the knife category, which means that the knife was a special category that the empty point cloud was classified as.

## G   Effects of the data augmentation on sensitivities

We explored effects of data augmentation on sensitivities in Section 4 in the paper, in this section in the supplementary material, we provide more details about the experimental setting and results. We conducted experiments to explore the effects of data augmentation on sensitivities, including (1) the translation data augmentation, (2) the scale data augmentation, (3) the rotation data augmentation around the y-axis, and (4) the rotation data augmentation around a random axis.

Table 2: Translation sensitivity of DNNs with or without the translation data augmentation.

| Network architecture | Model A: w/ translation data augmentation | Model B: w/o translation data augmentation |
|---|---|---|
| PointNet | 0.111± 0.053 | 0.160±0.067 |
| DGCNN | 0.048± 0.024 | 0.098±0.054 |

Table 3: Scale sensitivity of DNNs with or without the scale data augmentation.

| Network architecture | Model A: w/ the scale data augmentation | Model C: w/o the scale data augmentation |
|---|---|---|
| PointNet | 0.025±0.017 | 0.063±0.047 |
| DGCNN | 0.020±0.015 | 0.028±0.020 |

We conducted experiments on two network architectures, *i.e.,* PointNet and DGCNN. As Table 1 shows, for each network, we trained five versions of this network with different data augmentation settings. We compared model A and B to explore effects of the translation data augmentation. We compared model A and C to explore effects of the scale data augmentation. We compared model A and D to explore effects of the rotation data augmentation around the y-axis. We compared model A and E to explore effects of the rotation data augmentation around a random axis. These DNNs were trained on the ModelNet10 dataset. The baseline network (version A) was trained following the standard and widely-used setting in [4]. All the other versions B-E were revised from such a standard setting to enable fair comparisons.

We found that the translation data augmentation decreased the translation sensitivity of a DNN (as Table 2 shows), the scale data augmentation decreased the scale sensitivity of a DNN (as Table 3 shows), and the rotation data augmentation around the y-axis/a random axis decreased the rotation sensitivity of a DNN (as Tables 4 and 5 show).

## H  Comparing effects of the rotation data augmentation and the effects of adversarial training on the rotation sensitivity

In this section, we compared effects of the rotation data augmentation and the effects of adversarial training on the rotation sensitivity. We conducted experiments on DGCNN and GCNN. We trained two versions of DGCNN and GCNN, including Model E (see Table 1) and the adversarially-trained one (using rotations for attack).

Table 6 shows that compared with the rotation data augmentation, the adversarial training based on rotations of point clouds had a greater impact on the rotation sensitivity. *I.e.,* the adversarially-trained DNN had a lower value of rotation sensitivity than the DNN trained with the rotation data augmentation around a random axis.

## I  Effects of sizes of the selected regions on sensitivities

In this section, we conducted experiments to explore the effects of the size of the selected regions. In addition to the size used in our paper (each point cloud was partitioned to 32 regions), we selected other two sizes, *i.e.,* one size larger than the size in the paper and one size smaller than the size in the paper. To obtain regions with the larger size, each point cloud was partitioned to 16 regions. To obtain regions with the smaller size, each point cloud was partitioned to 64 regions. We measured the rotation sensitivity, the translation sensitivity, and the scale sensitivity of PointNet and DGCNN based on the ModelNet10 dataset, as shown in Table 7.

Table 4: Rotation sensitivity of DNNs with or without the rotation data augmentation around y-axis.

| Network architecture | Model D: w/ the rotation data augmentation around y-axis | Model A: w/o the rotation data augmentation around y-axis |
|---|---|---|
| PointNet | 0.108±0.047 | 0.155±0.068 |
| DGCNN | 0.158±0.063 | 0.173±0.072 |

Table 5: Rotation sensitivity of DNNs with or without the rotation data augmentation around a random axis.

| Network architecture | Model E: w/ the rotation data augmentation around a random axis | Model A: w/o the rotation data augmentation around a random axis |
|---|---|---|
| PointNet | 0.057±0.030 | 0.155±0.068 |
| DGCNN | 0.052±0.021 | 0.173±0.072 |

Table 6: Rotation sensitivity of DNNs with the rotation data augmentation and the adversarially-trained DNN.

| Model | Rotation sensitivity |
|---|---|
| Model E: DGCNN w/ rotation data augmentation around a random axis | 0.052 ±0.021 |
| Adversarially-trained DGCNN | 0.036±0.015 |
| Model E: GCNN w/ rotation data augmentation around a random axis | 0.048±0.024 |
| Adversarially-trained GCNN | 0.028±0.016 |

Results show that as the size of the selected regions increased, all sensitivities increased. However, the relative magnitude of sensitivities of different models did not change. We still obtained the same conclusion that all DNNs were more sensitive to the rotation than the translation and the scale change. Therefore, the observations and analysis in our paper are reliable.

Besides, we can also summarize the following two conclusions from Table 7, and these conclusions can also be verified by results in the paper. (1) DGCNN was more robust to the translation than the PointNet. (2) Both DGCNN and PointNet were robust to the scale change.

## J  Effects of the size of datasets on sensitivities

In this experiment, we aimed to explore the effects of the size of datasets on sensitivities. Specifically, we conducted experiments based on a larger dataset, *i.e.,* the ModelNet40 dataset, which had more training samples than the dataset used in our paper (*i.e.,* the ModelNet10 dataset). Experimental results show that the size of datasets did not significantly affect the metrics in most cases.

*Experiment settings:* Based on the ModelNet40 dataset, we trained two DNNs, *i.e.,* PointNet and DGCNN. We measured the rotation sensitivity, the translation sensitivity, and the scale sensitivity of these DNNs.

Table 8 shows that the relative relationship of each sensitivity between DNNs trained on the ModelNet40 dataset was the same as that trained on the ModelNet10 dataset, *i.e.,* all DNNs were more sensitive to the rotation than the translation and the scale change. This indicates that the conclusions in our paper are reliable.

Besides, we can also summarize the following two conclusions from Table 8, and these conclusions can also be verified by results in the paper. (1) DGCNN was more robust to the translation than the PointNet. (2) Both DGCNN and PointNet were robust to the scale change.

Table 7: Sensitivities of DNNs with different sizes of the selected regions.

| Model | region size | rotation sensitivity | translation sensitivity | scale sensitivity |
|---|---|---|---|---|
| PointNet | Small size (each point cloud was partitioned to 64 regions) | 0.086±0.044 | 0.061±0.031 | 0.013±0.011 |
| | Middle size (each point cloud was partitioned to 32 regions, in the paper) | 0.159±0.070 | 0.110±0.053 | 0.024±0.017 |
| | Large size (each point cloud was partitioned to 16 regions) | 0.292±0.117 | 0.202±0.095 | 0.043±0.029 |
| DGCNN | Small size (each point cloud was partitioned to 64 regions) | 0.091±0.043 | 0.026±0.013 | 0.012±0.008 |
| | Middle size (each point cloud was partitioned to 32 regions, in the paper) | 0.174±0.075 | 0.048±0.024 | 0.020±0.014 |
| | Large size (each point cloud was partitioned to 16 regions) | 0.336±0.144 | 0.086±0.047 | 0.035±0.027 |

Table 8: Sensitivities of DNNs trained on datasets with different sizes.

| Model | Dataset | rotation sensitivity | translation sensitivity | scale sensitivity |
|---|---|---|---|---|
| PointNet | ModelNet10 | 0.159±0.070 | 0.110±0.053 | 0.024±0.017 |
| | ModelNet40 | 0.162±0.073 | 0.069±0.035 | 0.026±0.023 |
| DGCNN | ModelNet10 | 0.174±0.075 | 0.048±0.024 | 0.020±0.014 |
| | ModelNet40 | 0.158±0.065 | 0.033±0.013 | 0.022±0.012 |

Table 9: Details about the training protocol of each DNN.

| Model | Optimizer | Epoch | Learning_rate | Scheduler |
|---|---|---|---|---|
| PointNet [2] —— following settings in [2] | Adam | 200 | 0.001 | StepLR |
| PointNet++ [3] —— following settings in [3] | Adam | 200 | 0.001 | StepLR |
| PointConv [6] —— following settings in [6] | SGD | 400 | 0.01 | StepLR |
| DGCNN [4] —— following settings in [4] | SGD | 250 | 0.1 | CosineAnnealingLR |
| GCNN [4] —— following settings in [4] | SGD | 250 | 0.1 | CosineAnnealingLR |

On the other hand, in most cases, DNNs with the same architecture trained on different datasets (with different numbers of training samples) had similar sensitivities. This indicates that the size of datasets had little impact on different metrics.

## K  Details about the training protocol of each DNN

In this section, we reported details about the training protocol of each DNN, as shown in Table 9. Actually, for all DNNs used in our paper, we strictly followed the training protocol in papers of these DNNs [2, 3, 6, 4].