# OpenReview forum: "Interpreting Representation Quality of DNNs for 3D Point Cloud Processing"
_NeurIPS.cc/2021/Conference — NeurIPS 2021 Poster_

### Official Review · Reviewer_qvpe · 2021-07-15

**Rating:** 6
**Confidence:** 5

**Summary:**

This paper measures the sensitivity of various point cloud classification networks (like PointNet, PointNet++, DGCNN) to rotation, translation, scale, and local 3D structure. It also proposes ways to measure the 3D smoothness of a network. Based on these analyses, the work provides various insights. One of the main insights being that the DNNs don’t handle rotation very well. Another insight is that training with adversarial perturbations helps.

**Limitations And Societal Impact:**

Please refer to weaknesses discussed in the main review.

**Main Review:**

Strengths

- The kind of analysis used in the paper to quantify different aspects of DNN could be useful for 3D point cloud processing.

Weakness

- The data augmentation could have a very significant impact on the conclusions in the paper (as explained below) but those details are missing. Specifically, it is unclear what kind of data augmentation strategies are used while training. For most of the best-performing models on point cloud classification, rotation augmentation is not used while training the DNN [1]. Only scale and translation augmentations are used. This is because while testing the object models are already axis-aligned. Hence, rotation augmentation does not help in improving the test set performance.  If this standard practice (i.e using scale and translation augmentation and not using rotation augmentation) is used in this paper as well, then the insights provided in the paper are very much expected, as a DNN would only be invariant to changes it has seen while training. Moreover, it would explain why adversarially trained GCNN, which would take rotated samples as input while training would be less sensitive to rotation. This would severely limit the contribution of the current work. Therefore without knowing this detail it would be difficult to judge the importance of the work.

- The paper lacks discussion about prior works which also point out that DNN for point cloud processing is sensitive to rotation (for example [1, 2]). The paper should add such a discussion to contextualize the findings with regards to what is already known.

- It would also be good to mention the accuracy of the DNNs used for the analysis to make sure that well-trained DNNs have been used. More details about the exact training protocol should also be provided.

- The resetting of absent points (L124-L132) could introduce unnecessary artifacts impacting the performance of DNNs. Most DNNs like PointNet, PointNet++, DGCNN should be able to handle the different number of points by default. It would be important to compare if simply removing points and replacing missing points with center lead to the same performance.

[1] Revisiting point cloud shape classification with a simple and effective baseline

[2] PointCNN: Convolution On -Transformed Points, NeuRIPS 2018

Minor Comments:

- It seems like many insights mentioned in the introduction repeat the same information. It would be useful to mention the insights in a more concise form. The reviewer feels that simpler and more fundamental insights would be more useful than many complicated ones; at least in the introduction.


Overall Recommendation:

In the current version, critical information is missing. Hence, one cannot judge how useful the contributions are. There are some other issues as well (see weak points). I look forward to the response by the authors. For now, I would recommend a borderline reject rating for the paper.


**Time Spent Reviewing:**

4

---

> ### Author Response · Authors · 2021-08-10
> **Response to Reviewer qvpe (Part1)**
>
> Thank you for your great efforts on the review of this paper. We will try our best to answer all your questions.
>
>
>
> Q1: About data augmentation settings used in this paper. Q1.1: Ask for details about the data augmentation strategies used in this paper. "The data augmentation could have a very significant impact on the conclusions in the paper (as explained below) but those details are missing. Specifically, it is unclear what kind of data augmentation strategies are used while training. ..." Q1.2: "If this standard practice (i.e using scale and translation augmentation and not using rotation augmentation) is used in this paper as well, then the insights provided in the paper are very much expected." Q1.3: The standard training without rotation data augmentation can "explain why adversarially trained GCNN, which would take rotated samples as input while training would be less sensitive to rotation." This would limit the contribution of the current work.
>
> A: Good questions. For Q1.1, in this paper, during the training process of each DNN, we used standard and widely-used data augmentations in [cite1], i.e., using scale and translation augmentation and not using rotation augmentation. We will clarify this in the final version.
>
> For Q1.2, our research is not limited to verifying a common phenomenon. Our metrics enable us to quantify utility of different data augmentations. To this end, we have **added experiments** to explore effects of data augmentation on sensitivities, including (1) the translation data augmentation, (2) the scale data augmentation, (3) the rotation data augmentation around the y-axis, and (4) the rotation data augmentation around a random axis.
>
> We conducted experiments on two network architectures, *i.e.,* PointNet and DGCNN. As Table 1 shows, for each network, we trained five versions of this network with different data augmentation settings. We compared model A and B to explore effects of the translation data augmentation. We compared model A and C to explore effects of the scale data augmentation. We compared model A and D to explore effects of the rotation data augmentation around the y-axis. We compared model A and E to explore effects of the rotation data augmentation around a random axis. Each DNN was trained on the ModelNet10 dataset. The baseline network (version A) was trained following the standard and widely-used setting in [cite1]. All the other versions B-E were revised from such a standard setting to enable fair comparisons.
>
> **Table 1: Different versions of a specific network architecture using different data augmentation.**
>
> | Model                           | translation data augmentation | scale data augmentation | rotation data augmentation around the y-axis | rotation data augmentation around a random axis |
> | ------------------------------- | ----------------------------- | ----------------------- | -------------------------------------------- | ----------------------------------------------- |
> | A: baseline settings in [cite1] | $\checkmark$                  | $\checkmark$            | $\times$                                     | $\times$                                        |
> | B                               | $\times$                      | $\checkmark$            | $\times$                                     | $\times$                                        |
> | C                               | $\checkmark$                  | $\times$                | $\times$                                     | $\times$                                        |
> | D                               | $\checkmark$                  | $\checkmark$            | $\checkmark$                                 | $\times$                                        |
> | E                               | $\checkmark$                  | $\checkmark$            | $\times$                                     | $\checkmark$                                    |
>
> We found that the translation data augmentation decreased the translation sensitivity of a DNN (as Table 2 shows), the scale data augmentation decreased the scale sensitivity of a DNN (as Table 3 shows), the rotation data augmentation around the y-axis/a random axis decreased the rotation sensitivity of a DNN (as Tables 4 and 5 show).
>
> **Table 2: Translation sensitivity of DNNs with or without the translation data augmentation.**
>
> | Network architecture | Model A: w/ the translation data augmentation | Model B: w/o the translation data augmentation |
> | :------------------: | --------------------------------------------- | ---------------------------------------------- |
> |       PointNet       | $0.111{\small\pm0.053}$​                       | $0.160{\small\pm0.067}$​                        |
> |        DGCNN         | $0.048{\small\pm0.024}$​                       | $0.098{\small\pm0.054}$​                        |
>
> **Table 3: Scale sensitivity of DNNs with or without the scale data augmentation**
>
> | Network architecture | Model A: w/ the scale data augmentation | Model C: w/o the scale data augmentation |
> | :------------------: | --------------------------------------- | ---------------------------------------- |
> |       PointNet       | $0.025{\small\pm0.017}$​                 | $0.063{\small\pm0.047}$​                  |
> |        DGCNN         | $0.020{\small\pm0.015}$​                 | $0.028{\small\pm0.020}$​                  |
>
> **Table 4: Rotation sensitivity of DNNs w/ or w/o the rotation data augmentation around y-axis**
>
> | Network architecture | Model D: w/ the rotation data augmentation around y-axis | Model A: w/o the rotation data augmentation around y-axis |
> | :------------------: | -------------------------------------------------------- | --------------------------------------------------------- |
> |       PointNet       | $0.108{\small\pm0.047}$​​                                  | $0.155{\small\pm0.068}$                                   |
> |        DGCNN         | $0.158{\small\pm0.063}$                                  | $0.173{\small\pm0.072}$                                   |
>
> **Table 5: Rotation sensitivity of DNNs w/ or w/o the rotation data augmentation around a random axis**
>
> | Network architecture | Model E: w/ the rotation data augmentation around a random axis | Model A: w/o the rotation data augmentation around a random axis |
> | :------------------: | ------------------------------------------------------------ | ------------------------------------------------------------ |
> |       PointNet       | $0.057{\small\pm0.030}$                                      | $0.155{\small\pm0.068}$                                      |
> |        DGCNN         | $0.052{\small\pm0.021}$                                      | $0.173{\small\pm0.072}$                                      |
>
> For Q1.3, although we agree with that learning without rotation augmentation can no doubt generate less rotation-robust model than adversarial training, we have conducted **new experiments** to further prove **a much stronger conclusion** that *adversarial training is more powerful than rotation augmentation, in terms of boosting rotation robustness*. To this end, we compare the effects of rotation data augmentation and the effects of adversarial training on the rotation sensitivity. We conducted experiments on GCNN. We trained four versions of GCNN, including model A (see Table 1), model D (see Table 1), Model E (see Table 1), and the adversarially-trained GCNN.
>
> Table 6 shows that compared with the rotation data augmentation, the adversarial training based on rotations of point clouds had a greater impact on the rotation sensitivity. *I.e.,* the adversarially-trained DNN had lower rotation sensitivity than the DNN with the rotation data augmentation around the y-axis and the DNN with the rotation data augmentation around a random axis.
>
> Therefore, our conclusion is convincing that adversarial training boosts the rotation robustness, no matter whether the baseline method has the rotation augmentation.
>
> **Table 6: Rotation sensitivity of DNNs with the rotation data augmentation and the adversarially-trained DNN**
>
> |                            Model                             | Rotation sensitivity    |
> | :----------------------------------------------------------: | ----------------------- |
> |        Model A: GCNN w/o  rotation data  augmentation        | $0.174{\small\pm0.066}$ |
> | Model D: GCNN w/  rotation data  augmentation around the y-axis | $0.147{\small\pm0.054}$ |
> | Model E: GCNN w/  rotation data  augmentation around a random axis | $0.048{\small\pm0.024}$ |
> |                **Adversarially-trained GCNN**                | $0.033{\small\pm0.011}$ |
>
> [cite1] DGCNN: Dynamic Graph CNN for Learning on Point Clouds. ACM Transactions on Graphics (TOG) 2019

---

> > ### Comment · Reviewer_qvpe · 2021-08-19
> > **Response**
> >
> > I would like to thank the authors for the detailed response and the additional experiments. I very much appreciate their effort.
> >
> > My concerns about missing discussion about prior works (weakness #2), information about whether the DNNs are well trained (weakness #3) and details about training protocol (weakness #4) have been addressed as the authors have promised to add these details in the next version of the paper.
> >
> > My minor concern about removing points and replacing them with the center has been addressed partially but I am not completely clear/convinced but the response in the rebuttal. Firstly, the rebuttal only mentions the limitations of PointNet++ and not about other architectures like PointNet and DGCNN. Second, even for PointNet++, it is not clear to me 512 is the theoretical limit for PointNet++. Especially because prior works like [1, 2] have evaluated performance for PointNet++ with a smaller number of points.
> >
> > [1] Relation-Shape Convolutional Neural Network for Point Cloud Analysis
> > [2] DensePoint: Learning Densely Contextual Representation for Efficient Point Cloud Processing
> >
> > However, I am very concerned regarding my primary concern (Weakness #1), i.e. the conclusions of the paper seem to be trivial when taking into account the data augmentation. As I earlier expected, the point-based models were sensitive to rotation as they were not trained with rotation augmentation. We can see that the sensitivity to rotation decreases significantly (Table 5 in rebuttal) when this augmentation is used. Hence, the introduction, conclusion, and insights in the paper (Insight 1, Insight 2, Insight 3, L36-45) should be revised significantly. In the reviewer's opinion this would be a significant change in the original paper.
> >
> > The rebuttal further argues that the new experiments "prove a much stronger conclusion that adversarial training is more powerful than rotation augmentation, in terms of boosting rotation robustness.". I am very convinced by this statement as in Table 6, the difference between "Model E: GCNN w/ rotation data augmentation around a random axis" and "Adversarially-trained GCNN" is not statistically significant. Hence, such a strong statement is not convincing given the evidence. I believe this should be further probed. It is also unclear why the performance is shown for only one model and why this comparison is not done for other architectures. I hope the authors would add numbers about other architectures as well.
> >
> > Hence, my primary concern that the conclusions in the paper might be trivial still holds. Therefore, I would like to keep the weak reject rating for the paper.

---

> > > ### Author Response · Authors · 2021-08-20
> > > **Response to further questions of Reviewer (qvpe) in the second round**
> > >
> > > Thank you very much for your further comments, and we will answer all your concerns.
> > >
> > > Q1: "The conclusions of the paper seem to be trivial when taking into account the data augmentation.  Hence, the introduction, conclusion, and insights in the paper (Insight 1, Insight 2, Insight 3, L36-45) should be revised significantly." "The rebuttal argues that new experiments prove a much stronger conclusion that adversarial training is more powerful than rotation augmentation, in terms of boosting rotation robustness. I believe this should be further probed and hope the authors would add numbers about other architectures as well."
> > >
> > > A1: Thank you for further clarifying your concerned issues, so that compared with the first round, we will give a more specific reply. We would like to answer this concern from the following two perspectives.
> > >
> > > First, the main contribution of this study is that this is the first attempt to explore **essential mechanisms** for several phenomena/insights in 3D point cloud processing (in particular, your mentioned phenomenon/insight that ***the DNN is sensitive to rotations***), **instead of the concerned proof of common insights**, to the best of our knowledge. In other words, compared with the conclusion/insight ***the DNN is sensitive to rotations***, exploring **the exact reason for such an insight is much more valuable**.
> > >
> > > To this end, our method also successfully explains the essential reasons for other phenomena from the perspectives of **six types of regional sensitivities, spatial smoothness, and representation complexity**.
> > >
> > > The **essential mechanism** for the mentioned rotation sensitivity has values beyond the proof of the rotation sensitivity.  Specifically, DNNs are sensitive to rotations, because most DNNs are not well trained to encode edges and corners, but at the same time, edges and corners are usually considered as important regions by DNNs (see Line 262-270). When we rotate an object to different angles, attributions of these rotation-sensitive regions may be either significantly positive or significantly negative (see Fig. 4 (a) ). Therefore, this research **reveals these DNNs' essential representation flaws in a fine-grained manner** (*i.e.*, DNNs mistakenly consider edges and corners important but are unable to encode them well), which may guide the design of DNNs in the future.
> > >
> > > Besides, the reason for the robustness ensured by adversarial training has been novelly explained as the increase of the extremely high-order interactions in terms of representation complexity.
> > >
> > > Second, the difference between rotation augmentation and adversarial training is statistically significant. It is easy to reduce the rotation sensitivity from 0.174 to 0.048. However, it is difficult to further reduce the rotation sensitivity from 0.048 to 0.033, which essentially distinguishes robust models and non-robust models. We are conducting **new experiments** to compare the rotation robustness between models with rotation augmentation and adversarially trained models. Please wait. Thank you.
> > >
> > > ---
> > >
> > > Q2. "My minor concern about removing points and replacing them with the center has been addressed partially but I am not completely clear/convinced but the response in the rebuttal."
> > > Q2.1: About limitations for “other architectures like PointNet and DGCNN.”
> > > Q2.2: “It is not clear to me 512 is the theoretical limit for PointNet++. " "Prior works like [cite1, cite2] have evaluated performance for PointNet++ with a smaller number of points.”
> > >
> > > A2: Thank you for further clarifying your concerned issues.
> > >
> > > For Q2.1, let us explain why the number of input points must not be too small for other two architectures (DGCNN and PointNet). First, for DGCNN, if the number of input points is less than 20 points, the k-nearest neighbor operation cannot be conducted, and DGCNN crashes. Note that the computation of Shapley values requires to make inference on a point cloud fragment with a single region $i$​ (*i.e.*, $x_{\\{i\\}}$​​​​), which may have less than 20 points. In this case, DGCNN cannot work. Second, PointNet is able to handle different numbers of input points. However, for a fair comparison, we uniformly replaced points with center for all DNNs.
> > >
> > > For Q2.2, prior works [cite1, cite2] actually used 1024 input points, rather than a smaller number of input points. The feeling of these works using fewer points is because of the confusing statements in these papers. They actually reset some points to a specific position, just like what we did, instead of using fewer points.
> > >
> > > [cite1] Relation-Shape Convolutional Neural Network for Point Cloud Analysis
> > >
> > > [cite2] DensePoint: Learning Densely Contextual Representation for Efficient Point Cloud Processing
> > >
> > >
> > > If there are still some concerns, we welcome your further comments and expect more discussions.

---

> > > > ### Comment · Reviewer_qvpe · 2021-08-28
> > > > **Response**
> > > >
> > > > I would like to thank the authors for their sincere effort and conducting additional experiments.
> > > >
> > > > I agree with the authors that the primary contribution is in the mechanism to explain the phenomenon and necessarily the particular insights. However, it is necessary that the insights are correct and do not propagate incorrect information/knowledge in the community and literature. It is from this point of view, the paper needs to be significantly updated from the current version as it seems to suggest in the Introduction (L36-45, L66-69) and Conclusion (L304-306) that DNNs are **extremely sensitive to rotation** (L307). This is not the case as demonstrated by the experiments here and the sensitivity is largely affected by the data augmentation used.
> > > >
> > > > Surprisingly, the authors say "Compared with the rotation sensitivity of DGCNN with rotation augmentation, the rotation sensitivity of adversarially trained DGCNN is reduced from 0.052 to 0.036. Compared with the rotation sensitivity of GCNN with rotation augmentation, the rotation sensitivity of adversarially trained GCNN is reduced from 0.048 to 0.028. Such reduction is statistically significant." Statistical significance is measure by taking into account mean **as well as** the variance. And as the authors have reported the rotation sensitivity for adversarially trained GCNN is **0.033 +- 0.011** while that for rotation augmented GCNN is **0.048 +- 0.024**. The difference between the two is not statistically significant by any acceptable measure of statistically significance. The authors could conduct more trials/runs to establish statistical significance. Finally, the readers would appreciate it if similar variance measures are measured for DGCNN as well.
> > > >
> > > > To summarize, some specific changes that would be useful are:
> > > >
> > > > - Update the Insights (Introduction and Conclusion) to reflect the results of the new experiments
> > > > - Report the additional experiments with rotation sensitivity
> > > > - Discuss the prior works related to rotation sensitivity
> > > >
> > > > As the authors have promised to make these amendments, I feel more comfortable with the paper and would like to update my rating.
> > > >
> > > > I look forward to the updated version of the paper and wish the authors the best of luck!

---

> > > > > ### Author Response · Authors · 2021-08-29
> > > > > **Response to Reviewer (qvpe)**
> > > > >
> > > > > Thank you very much for your great efforts on the review of this paper. We will follow your suggestions to update the paper.

---

> > > ### Author Response · Authors · 2021-08-26
> > > **Response to further questions of Reviewer (qvpe) in the second round (Part2)**
> > >
> > > To prove that the difference between the rotation augmentation and the adversarial training is statistically significant, we have conducted **new experiments** to compare the rotation robustness between models with the rotation augmentation and adversarially trained models. We have conducted experiments on two architectures, DGCNN and GCNN. Note that we increased the number of training epochs for all models, so that all models were more converged compared to those in our first-round rebuttal.
> > >
> > > | Model | rotation augmentation | adversarial training |
> > > | ----- | --------------------- | -------------------- |
> > > | DGCNN | 0.052                 | 0.036                |
> > > | GCNN  | 0.048                 | 0.028                |
> > >
> > > The above table shows the rotation sensitivity of different models. Results prove that compared with the rotation augmentation, the adversarial training based on rotations had a more statistically significant impact on the rotation sensitivity. Compared with the rotation sensitivity of DGCNN with rotation augmentation, the rotation sensitivity of adversarially trained DGCNN is reduced from 0.052 to 0.036. Compared with the rotation sensitivity of GCNN with rotation augmentation, the rotation sensitivity of adversarially trained GCNN is reduced from 0.048 to 0.028. Such reduction is statistically significant.
> > >
> > > The objective of this study is not limited to the proof of several phenomena/insights in 3D point cloud processing (in particular, the adversarial training increases rotation robustness). Besides, the more important goal is to explore essential mechanisms for these phenomena/insights. Specifically, the reason for the robustness ensured by adversarial training has been explained as the increase of the extremely high-order interactions. Previous experimental results have proved the above essential mechanisms. Nevertheless, results of new experiments further proved that the difference between the rotation augmentation and the adversarial training is statistically significant.

---

> ### Author Response · Authors · 2021-08-10
> **Response to Reviewer qvpe (Part2)**
>
> Q2: Discussions about prior works [cite2, cite3], “The paper lacks discussion about prior works which also point out that DNN for point cloud processing is sensitive to rotation (for example [cite2, cite3]). The paper should add such a discussion to contextualize the findings with regards to what is already known.”
>
> A: ﻿ Thanks. PointCNN [cite3] is a classical point cloud processing model, which discovers that a large portion of the 3D models from ModelNet40 are pre-aligned to the common up direction and horizontal facing direction. [cite2] discovers that rotation augmentation adversely affects the performance of a model. One of our insights "DNNs for point cloud processing are sensitive to rotation" is consistent with the findings of [cite2, cite3].
>
> However, instead of focusing on improving classification performance  [cite2, cite3], our study proposed a generic method to analyze the representation quality of DNNs for 3D point cloud processing, as well as the quality of regional representations. Nevertheless, we will cite these two insightful papers in our final version and add discussions about their findings.
>
> [cite2] Revisiting point cloud shape classification with a simple and effective baseline
>
> [cite3] PointCNN: Convolution On -Transformed Points, NeuRIPS 2018
>
> ---
>
> Q3: Ask for the accuracy of DNNs used in our paper. “It would also be good to mention the accuracy of the DNNs used for the analysis to make sure that well-trained DNNs have been used.”
>
> A: ﻿ Thanks. All DNNs used in our paper were well-trained. We have followed your suggestions to report the accuracy of DNNs used in our paper, as follows. PointNet, 93.5%, PointNet++, 94.7%, DGCNN, 94.4%, GCNN, 95.1%, PointConv, 94.6%, and adversarially-trained GCNN, 91.0% on ModelNet10. We also reported the accuracy of other DNNs trained on the same dataset (i.e., ModelNet10 dataset), e.g., RotationNet [cite14], 98.46%, SPNet[cite4], 97.25%, SO-Net[cite5], 95.7%, 3DmFV-Net[cite6], 95.2%, MHBN[cite7], 95%, 3DCapsule[cite8] 94.7%, KCNet[cite9], 94.4%, FoldingNet[cite10], 94.4%, Pairwise[cite11], 92.8%, GIFT[cite12], 92.35%, ECC[cite13], 90.0%. Considering the accuracy distribution of these DNNs, the DNNs used in our paper are well-trained. We will report the accuracy of DNNs used in our paper in our final version.
>
> [cite4] Yavartanoo et al. SPNet: Deep 3D Object Classification and Retrieval using Stereographic Projection, ACCV2018.
>
> [cite5] Li et al. SO-Net: Self-Organizing Network for Point Cloud Analysis. CVPR2018
>
> [cite6] Ben-Shabat et al. 3D Point Cloud Classification and Segmentation using 3D Modified Fisher Vector Representation for Convolutional Neural Networks arXiv 2017.
>
> [cite7] Yu et al. Multi-view Harmonized Bilinear Network for 3D Object Recognition. CVPR 2018.
>
> [cite8] A. Cheraghian and L. Petersson, 3DCapsule: Extending the Capsule Architecture to Classify 3D Point Clouds, 2019 IEEE Winter Conference on Applications of Computer Vision (WACV), 2019, pp. 1194-1202.
>
> [cite9] Shen et al. Mining Point Cloud Local Structures by Kernel Correlation and Graph Pooling. CVPR 2018
>
> [cite10] Yang et al. FoldingNet: Point Cloud Auto-encoder via Deep Grid Deformation. CVPR 2018
>
> [cite11] Edward Johns, Stefan Leutenegger and Andrew J. Davison. Pairwise Decomposition of Image Sequences for Active Multi-View Recognition CVPR 2016.
>
> [cite12] Bai et al. Longin Jan Latecki. GIFT: A Real-time and Scalable 3D Shape Search Engine. CVPR 2016.
>
> [cite13] Martin Simonovsky, Nikos Komodakis Dynamic Edge-Conditioned Filters in Convolutional Neural Networks on Graphs.
>
> [cite14] Kanezaki et al. RotationNet: Joint Object Categorization and Pose Estimation Using Multiviews from Unsupervised Viewpoints. CVPR, 2018.
>
> ---
>
> Q4: “More details about the exact training protocol should also be provided.”
>
> A: ﻿ Thanks. We have followed your suggestions to show details about the training protocol of each DNN as follows. Actually, for all DNNs used in our paper, we strictly followed the training protocol in papers of these DNNs[cite18, 15, 16, 17]. We will add these details in our final version.
>
> |                                                         | Optimizer | Epoch | Learning_rate | scheduler         |
> | ------------------------------------------------------- | --------- | ----- | ------------- | ----------------- |
> | PointNet [cite18] $--$​​​​ following settings in [cite18]   | Adam      | 200   | 0.001         | StepLR            |
> | PointNet++ [cite15] $--$​​​​ following settings in [cite15] | Adam      | 200   | 0.001         | StepLR            |
> | DGCNN  [cite16] $--$​​​​ following settings in [cite16]     | SGD       | 250   | 0.1           | CosineAnnealingLR |
> | GCNN  [cite16] $--$​​​​ following settings in [cite16]      | SGD       | 250   | 0.1           | CosineAnnealingLR |
> | PointConv  [cite17] $--$​​​​ following settings in [cite17] | Adam      | 400   | 0.001         | StepLR            |
>
> [cite15] PointNet++: Deep Hierarchical Feature Learning on Point Sets in a Metric Space, NeurIPS 2018
>
> [cite16] DGCNN: Dynamic Graph CNN for Learning on Point Clouds ACM. Transactions on Graphics (TOG) 2019
>
> [cite17] PointConv: Deep Convolutional Networks on 3D Point Clouds, CVPR 2019
>
> [cite18] PointNet: Deep Learning on Point Sets for 3D Classification and Segmentation. CVPR 2017
>
> ---
>
> Q5: “The resetting of absent points (L124-L132) could introduce unnecessary artifacts impacting the performance of DNNs. Most DNNs like PointNet, PointNet++, DGCNN should be able to handle the different number of points by default. It would be important to compare if simply removing points and replacing missing points with center lead to the same performance.”
>
> A: ﻿ A good suggestion, but we have proved that there is a theoretical flaw which hampers the suggested operation of removing points. Theoretically, we have proved that the number of input points must be greater than the number of points required for the sampling operation at the first layer of PointNet++, *i.e.,* 512 points. It is because the sampling operation at the first layer of PointNet++ requires the input points to be more than the 512 sampling points. Otherwise, the sampling operation cannot be conducted. Note that when we compute the Shapley value of a point cloud region, we need to input a point cloud with a single region, which usually has less than 512 points. Therefore, the suggested method does not work in this case. In this way, we choose the alternative plan, *i.e.,* replacing missing points with center for all DNNs.
>
> ---
>
> Q6: About paper writing. “It seems like many insights mentioned in the introduction repeat the same information. It would be useful to mention the insights in a more concise form. The reviewer feels that simpler and more fundamental insights would be more useful than many complicated ones; at least in the introduction.”
>
> A: ﻿ Thanks. We will follow your suggestions to polish our language to make the insights more concise.

---

### Official Review · Reviewer_NChg · 2021-07-18

**Rating:** 7
**Confidence:** 4

**Summary:**

This paper studies how to understand the quality of knowledge representations encoded in point cloud DNNs. Based upon the Shapley value, the paper presents six metrics to evaluate the model vulnerability toward different types of input variations. In addition, it presents an analysis regarding the smoothness and multi-order interactions of these attributions, pointing out the restrictions of current point DNNs. The paper makes interesting observations and draws useful conclusions, supported by a range of experimental evaluations.

**Limitations And Societal Impact:**

As I mentioned, there are some details unclear. Addressing them could potentially strengthen the submission. It would be interesting to include sparse voxel conv based methods given that it is one of the main streams for sparse point cloud analysis. It would also be interesting to see whether rotation equivariant networks could mitigate the rotation sensitivity issue.

**Main Review:**

This is a diagnosis paper studying an important problem. The submission is clearly written, making it easy to follow. The content is also well organized with enough technical details provided. Generally speaking, the submission is technically sound. The motivation behind different metric designs is clear and the experimental results successfully support the main claims. The diagnosis is quite new for 3D point cloud analysis. Its difference from previous works is also clearly discussed. The related work is adequate as far as I can tell.

I still have some questions and I hope the authors could provide more explanations to strengthen the submission.
1. Is the conclusions task dependent? Would the observation change if analyzing with another task, say semantic segmentation?
2. Is the smoothness score defined for a specific transformation or averaged across all types of transformations considered?
3. Is there any intuition why does the biased knife category appears for adversarial training?
4. Why does the GCNN mainly encodes extremely low-order or high-order interactions but not something in between?
5. How does the size of the selected regions influence the observations and analysis?

I think the paper is making an interesting analysis which reveals the limitations of current point DNNs and could provide intuitions for future designs.

**Time Spent Reviewing:**

3

---

> ### Author Response · Authors · 2021-08-10
> **Response to Reviewer NChg (Part1)**
>
>
> Thank you for your great efforts on the review of this paper. We will try our best to answer all your questions.
>
>
>
> Q1: “Is the conclusions task dependent? Would the observation change if analyzing with another task, say semantic segmentation?”
>
> A: ﻿ Thanks. The conclusions in our study are valid for the classification task. Actually, different tasks learn features with different properties. We have followed your suggestions to **add new experiments** on the reconstruction task. We conducted experiments on an AutoEncoder, which used all layers before the fully-connected layers in PointNet [cite1] as the encoder and used the decoder in the FoldingNet [cite2] as the decoder. We took $v(S)=\frac{(z(N)-z(\emptyset))^{\top}(z(S)-z(\emptyset))}{||z(N)-z(\emptyset)||_2}$​ as the reward score in Equation (1), where $z(S)$​ denotes the vector of latent representation given the input point cloud only containing regions in $S$​. This reward score measures the utility of the latent representation given regions in $S$​​ along the direction of the representation given the entire point cloud.
>
> We measured the rotation sensitivity, the translation sensitivity, and the scale sensitivity of the AutoEncoder, as follows. We found that (1) both the DNN for classification (except for the adversarially-trained GCNN) and the AutoEncoder for reconstruction were sensitive to rotation; (2) the AutoEncoder for reconstruction was more sensitive to translation than the DNN for classification; (3) the AutoEncoder for reconstruction was sensitive to the scale change.
>
> **Table 1: Comparison of sensitivities of DNNs for the classification task and sensitivities of an Autoencoder for the reconstruction task**
>
> | DNN                                           | rotation  sensitivity   | translation  sensitivity  | scale  sensitivity      |
> | --------------------------------------------- | ----------------------- | ------------------------- | ----------------------- |
> | PointNet for classification                   | $0.159{\small\pm0.070}$​ | $0.110{\small\pm0.053}$​   | $0.024{\small\pm0.017}$ |
> | PointNet++ for classification                 | $0.171{\small\pm0.064}$​ | $0.004{\small\pm0.004}$   | $0.054{\small\pm0.027}$ |
> | PointConv for classification                  | $0.145{\small\pm0.060}$ | $2.3e-4{\small\pm1.9e-4}$ | $0.027{\small\pm0.019}$ |
> | DGCNN for classification                      | $0.174{\small\pm0.075}$ | $0.048{\small\pm0.024}$   | $0.020{\small\pm0.014}$ |
> | GCNN for classification                       | $0.174{\small\pm0.067}$ | $0.050{\small\pm0.026}$   | $0.020{\small\pm0.014}$ |
> | adversarially-trained GCNN for classification | $0.034{\small\pm0.012}$ | $0.007{\small\pm0.004}$   | $0.020{\small\pm0.014}$ |
> | AutoEncoder for reconstruction                | $0.099{\small\pm0.043}$​ | $0.152{\small\pm0.086}$​   | $0.045{\small\pm0.042}$​ |
>
> [cite1] PointNet: Deep Learning on Point Sets for 3D Classification and Segmentation. CVPR 2017
>
> [cite2] Yang et al. FoldingNet: Point Cloud Auto-encoder via Deep Grid Deformation. CVPR 2018
>
> ---
>
> Q2: “Is the smoothness score defined for a specific transformation or averaged across all types of transformations considered?”
>
> A: Thanks. We defined two types of smoothness scores based on two types of transformation $T$ in Equation (5). One was defined for the rotation transformation, and the other was defined for the translation transformation, as follows.
>
> $$
> \textit{non-smoothness}^{\rm translation} = \mathbb{E}\_{x\in X}\mathbb{E}\_{T\_{\rm translation}}\mathbb{E}\_i\mathbb{E}\_{j\in\mathcal{N}(i)}\Big[\frac{ |   \phi\_{x'}(i)-\phi\_{x'}(j)|  }{Z\_{\rm smooth}} \Big| \_{x'=T\_{\rm translation}(x)} \Big]
> $$
>
> $$
> \textit{non-smoothness}^{\rm rotation} = \mathbb{E}\_{x\in X}\mathbb{E}\_{T_{\rm rotation}}\mathbb{E}\_i\mathbb{E}\_{j\in\mathcal{N}(i)}\Big[\frac{ |   \phi\_{x'}(i)-\phi\_{x'}(j)|  }{Z\_{\rm smooth}} \Big| \_{x'=T\_{\rm rotation}(x)} \Big]
> $$
>
> ---
>
> Q3: “Is there any intuition why does the biased knife category appears for adversarial training?”
>
> A: Thanks. We have followed your suggestion to **add experiments** to explore whether this phenomenon (biased knife category) is a specific situation or an accident. Specifically, we explore the bias appearing in the adversarially-trained DNNs with different initial parameters. If the bias of different DNNs is always the same (*i.e.,* all DNNs classify an empty input as the knife category), then it means that the knife is a special category that the empty point cloud is classified as. If the bias of different DNNs is different, then it means that this phenomenon is an accident.
>
> We trained the adversarially-trained GCNN five times under different initial parameters. The bias appearing in different DNNs was always the knife category, which means that the knife was a special category that the empty point cloud was classified as.

---

> ### Author Response · Authors · 2021-08-10
> **Response to Reviewer NChg (Part2)**
>
> Q4: “Why does the GCNN mainly encodes extremely low-order or high-order interactions but not something in between?”
>
> A: A good question. We obtained this conclusion through experimental observations of two adversarially-trained GCNNs trained  on the ModelNet10 dataset and the ShapeNet dataset. This phenomenon is very salient in adversarially-trained GCNN. Based on these observations, we will further explore the reason behind this phenomenon in the future study.
>
>  ---
>
>
> Q5: “How does the size of the selected regions influence the observations and analysis?”
>
> A: Thanks. We have conducted **new experiments** to explore the effects of the size of the selected regions. In addition to the size used in our paper (each point cloud was partitioned to 32 regions), we selected other two sizes, *i.e.,* one size larger than the size in the paper and one size smaller than the size in the paper. To obtain regions with the larger size, each point cloud was partitioned to 16 regions. To obtain regions with the smaller size, each point cloud was partitioned to 64 regions. We measured the rotation sensitivity, the translation sensitivity, and the scale sensitivity of PointNet and DGCNN based on the ModelNet10 dataset, as follows.
>
> Results show that as the size of the selected regions increased, all sensitivities will increased. However, the relative magnitude of sensitivities of different models did not change. We still obtained the same conclusion that all DNNs were more sensitive to the rotation than the translation and the scale change. Therefore, the observations and analysis in our paper are reliable.
>
> Besides, we can also summarize the following two conclusions from the following tables, and these conclusions can also be verified by results in the paper. (1) DGCNN was more robust to the translation than the PointNet. (2) Both DGCNN and PointNet were robust to the scale change.
>
> **Table 2: Comparison of sensitivities of PointNet with different sizes of the selected regions**
>
> |                                                              | rotation  sensitivity   | translation  sensitivity | scale  sensitivity      |
> | ------------------------------------------------------------ | ----------------------- | ------------------------ | ----------------------- |
> | Small  size (each point cloud was partitioned to 64 regions) | $0.086{\small\pm0.044}$​​​ | $0.061{\small\pm0.031}$​​​  | $0.013{\small\pm0.011}$​​​ |
> | Middle  size (each point cloud was partitioned to 32 regions, in our paper) | $0.159{\small\pm0.070}$​​​ | $0.110{\small\pm0.053}$​  | $0.024{\small\pm0.017}$​ |
> | Large  size (each point cloud was partitioned to 16 regions) | $0.292{\small\pm0.117}$​​​​ | $0.202{\small\pm0.095}$​​​​​  | $0.043{\small\pm0.029}$​​​​ |
>
> **Table 3: Comparison of sensitivities of DGCNN with different sizes of the selected regions**
>
> |                                                              | rotation  sensitivity   | translation  sensitivity | scale  sensitivity      |
> | ------------------------------------------------------------ | ----------------------- | ------------------------ | ----------------------- |
> | Small  size (each point cloud was partitioned to 64 regions) | $0.091{\small\pm0.043}$​ | $0.026{\small\pm0.013}$​  | $0.012{\small\pm0.008}$​ |
> | Middle  size (each point cloud was partitioned to 32 regions, in our paper) | $0.174{\small\pm0.075}$​ | $0.048{\small\pm0.024}$  | $0.020{\small\pm0.014}$​ |
> | Large  size (each point cloud was partitioned to 16 regions) | $0.336{\small\pm0.144}$​ | $0.086{\small\pm0.047}$​  | $0.035{\small\pm0.027}$​ |
>
> ---
>
>
> Q6: “It would also be interesting to see whether rotation equivariant networks could mitigate the rotation sensitivity issue.”
>
> A: Thanks. We have followed your suggestions to explore whether rotation equivariant networks could mitigate the rotation sensitivity issue. We measured the rotation sensitivity of two rotation-equivariant DNNs for 3D point cloud processing [cite4]. Each rotation-equivariant DNN was trained on the ModelNet10 dataset. We measured the rotation sensitivity on these two DNNs, as follows. We found that compared with traditional DNNs (even the adversarially-trained GCNN) used in our paper, rotation-equivariant DNNs were much more robust to the rotation.
>
> **Table 7: Comparison of sensitivities of DNNs used in our paper and rotation-equivariant DNNs [cite4]**
>
> | DNNs  used in our paper          | rotation  sensitivity   |
> | -------------------------------- | ----------------------- |
> | PointNet                         | $0.159{\small\pm0.070}$​ |
> | PointNet++                       | $0.171{\small\pm0.064}$​ |
> | PointConv                        | $0.145{\small\pm0.060}$ |
> | DGCNN                            | $0.174{\small\pm0.075}$ |
> | GCNN                             | $0.174{\small\pm0.067}$ |
> | Adversarially-trained GCNN       | $0.034{\small\pm0.012}$ |
> | Rotation-equivariant  PointNet++ | $0.002{\small\pm0.001}$ |
> | Rotation-equivariant  DGCNN      | $0.010{\small\pm0.004}$​​​​ |
>
> [cite4] Shen et al. 3D-Rotation-Equivariant Quaternion Neural Networks. In ECCV 2020.

---

> > ### Comment · Reviewer_NChg · 2021-08-23
> > **Reviewer's Response**
> >
> > Thanks for addressing my concerns and I would like to keep my original rating.

---

### Official Review · Reviewer_xqQ9 · 2021-07-23

**Rating:** 7
**Confidence:** 5

**Summary:**

This paper introduces metrics that reflect the representation quality, and properties, acquired by a deep-net that is applied in 3D Point-Cloud (PC) processing tasks. Concretely, the authors introduce six novel metrics that in aggregate capture different types of regional sensitivities e.g., the sensitivity of the DNN in rotations on specific regions/rotation-angles of the input PC. They further explore the extent that modern architectures in standard benchmarks acquire “spatial smoothness” --  do nearby points have ‘similar’ effects/importance for the end task? And finally, they explore the acquired representation complexity of the DNN by adapting a 'multi-order interaction' metric to PC data.

These metrics are being applied on a well-selected subset of modern PC-DNN architectures such PointNet++ or DGCNN when these are trained either for PC-object classification, or part-based object-segmentation. The analysis of the metrics in these tasks/architectures highlights a number of interesting points that oftentimes are very simple/intuitive: e.g., PointNet fails to encode local 3D structures (e.g., compared to PointNet++) — or includes results that are more subtle, but still highly intuitive & rich: e.g., adversarially training a PC-DNN with rotation/translation attacks; will force it to learn more global PC structures compared to standard training; which is intuitive as the local structures, per the new metrics, are those that on average are more susceptible to rotations.

**Ethical Concerns:**

I do not have any ethical concerns about this paper.

**Limitations And Societal Impact:**

Yes.

**Main Review:**

This paper is original, written with clarity, and introduces ideas, per the evaluation metrics that I think will be very helpful for the PC-Deep-Learning community. The exposition is clear and most findings well align with intuitions known by practitioners.  For instance, it is widely known that PointNet based variants do not cope well with rotations and that ad-hoc mechanisms that are typically applied to address this issue (e.g., the pc/feature rotation transformation loss in the original PointNet) have very limited success -- which is partially why a lot of work keeps happening in rotation-invariant or equivariant PC-DNNs. This paper brings a very elegant instrument to measure such a dependence (among others): the Shapley values. These can capture the sensitivity of different PC "sub-regions" to a specific quantity of interest, not only rotations but other structures like edge vs. surface vs. volume like structures. To this end, another interesting finding is that PointNet++ is the network that pays most attention to intrinsic 3D structures: "masses", which can be why in many settings is still the better performing, e.g., see [1].

The findings regarding how the adversarially trained net shifted its attention from orientations and positions to structural information and increased its spatial smoothness are important and they demonstrate a use case of how the new metrics can be used to examine/analyze the behavior of black-box models, training losses, etc.

The only concern that I have with the paper is that IMO lacks extensive use-cases analysis. While the networks architectures tested seem adequate; the datasets/tasks are less extensive. E.g., why not using ShapeNet  (or ModelNet50) for a larger classification problem? How is the training size affects the different metrics? How about trying auto-encoders or other forms of learning used for point-cloud processing? While, this seems like an extra mile that another paper can do; if you had included analysis say on AutoEncoders, GANs etc. to complete the picture, I would argue this paper should have very strong acceptance.


Minor:
L187. is represented ->  are
Figure 3. (b) -> are these objects supposed to be at different angles? I cannot tell.


[1] Goyal et al. Revisiting Point Cloud Shape Classification with a Simple and Effective Baseline





**Time Spent Reviewing:**

6

---

> ### Author Response · Authors · 2021-08-10
> **Response to Reviewer xqQ9**
>
>  Thank you for your great efforts on the review of this paper. We will try our best to answer all your questions.
>
>
>
> Q1: About extensive analysis for different datasets and tasks. “While the networks architectures tested seem adequate; the datasets/tasks are less extensive.”
>
> Q1.1: “*E.g.,* why not using ShapeNet (or ModelNet50) for a larger classification problem? How is the training size affects the different metrics?”
>
> Q1.2: “How about trying auto-encoders or other forms of learning used for point-cloud processing? … if you had included analysis say on AutoEncoders, GANs etc. to complete the picture, I would argue this paper should have very strong acceptance.”
>
>  A: Thank you. We have followed your suggestions to **add additional experiments** on different datasets and tasks.
>
> For Q1.1, we have added additional experiments based on a larger dataset, *i.e.,* the ModelNet40 dataset, which had larger *training size* than the dataset used in our paper (*i.e.,* ModelNet10 dataset). These experiments show that the *training size* does not significantly affect the metrics in most cases.
>
> *Experiment settings:* Based on the ModelNet40 dataset, we trained two DNNs, *i.e.*, PointNet and DGCNN. We measured the rotation sensitivity, the translation sensitivity, and the scale sensitivity of these DNNs.
>
> Results show that the relative relationship of each sensitivity between DNNs trained on the ModelNet40 dataset was the same as that trained on the ModelNet10 dataset, *i.e.,* all DNNs were more sensitive to the rotation than the translation and the scale change. This indicates that the conclusions in our paper are reliable.
>
> Besides, we can also summarize the following two conclusions from the following tables, and these conclusions can also be verified by results in the paper. (1) DGCNN was more robust to the translation than the PointNet. (2) Both DGCNN and PointNet were robust to the scale change.
>
> On the other hand, in most cases, DNNs with the same architecture trained on different datasets (with different *training size*) had similar sensitivities. This indicates that the *training size* has little impact on different metrics.
>
> **Table 1: Comparison of sensitivities of PointNet based on different datasets (with different *training size*)**
>
> | Dataset                                          | rotation  sensitivity   | translation  sensitivity | scale  sensitivty       |
> | ------------------------------------------------ | ----------------------- | ------------------------ | ----------------------- |
> | ModelNet10 (small *training size*, in our paper) | $0.159{\small\pm0.070}$ | $0.110{\small\pm0.053}$  | $0.024{\small\pm0.017}$ |
> | ModelNet40 (large *training size*)               | $0.162{\small\pm0.073}$ | $0.069{\small\pm0.035}$  | $0.026{\small\pm0.023}$ |
>
> **Table 2: Comparison of sensitivities of DGCNN based on different datasets (with different *training size*)**
>
> | Dataset                                          | rotation  sensitivity   | translation  sensitivity | scale  sensitivity      |
> | ------------------------------------------------ | ----------------------- | ------------------------ | ----------------------- |
> | ModelNet10 (small *training size*, in our paper) | $0.174{\small\pm0.075}$ | $0.048{\small\pm0.024}$  | $0.020{\small\pm0.014}$ |
> | ModelNet40 (large *training size*)               | $0.158{\small\pm0.065}$ | $0.033{\small\pm0.013}$  | $0.022{\small\pm0.012}$ |
>
> For Q1.2, we have added additional experiments based on an AutoEncoder for the reconstruction task, which used all layers before the fully-connected layers in PointNet [cite1] as the encoder and used the decoder in the FoldingNet [cite2] as the decoder. We took $v(S)=\frac{(z(N)-z(\emptyset))^{\top}(z(S)-z(\emptyset))}{||z(N)-z(\emptyset)||_2}$ as the reward score in Equation (1), where $z(S)$ denotes the vector of latent representation given the input point cloud only containing regions in $S$. This reward score measures the utility of the latent representation given regions in $S$ along the direction of the representation given the entire point cloud.
>
> We measured the rotation sensitivity, the translation sensitivity, and the scale sensitivity of these DNNs. We found that (1) both the DNN for classification (except for the adversarially-trained GCNN) and the AutoEncoder for reconstruction were sensitive to rotation; (2) the AutoEncoder for reconstruction was more sensitive to translation than the DNN for classification; (3) the AutoEncoder for reconstruction was sensitive to the scale change.
>
> **Table 3: Comparison of sensitivities of DNNs for the classification task and sensitivities of an Autoencoder for the reconstruction task**
>
> | DNN      | rotation  sensitivity | translation  sensitivity | scale  sensitivity |
> | -------- | --------------------- | ------------------------ | ------------------ |
> | PointNet for classification                   | $0.159{\small\pm0.070}$​ | $0.110{\small\pm0.053}$​   | $0.024{\small\pm0.017}$ |
> | PointNet++ for classification                 | $0.171{\small\pm0.064}$​ | $0.004{\small\pm0.004}$   | $0.054{\small\pm0.027}$ |
> | PointConv for classification                  | $0.145{\small\pm0.060}$ | $2.3e-4{\small\pm1.9e-4}$ | $0.027{\small\pm0.019}$ |
> | DGCNN for classification                      | $0.174{\small\pm0.075}$ | $0.048{\small\pm0.024}$   | $0.020{\small\pm0.014}$ |
> | GCNN for classification                       | $0.174{\small\pm0.067}$ | $0.050{\small\pm0.026}$   | $0.020{\small\pm0.014}$ |
> | adversarially-trained GCNN for classification | $0.034{\small\pm0.012}$ | $0.007{\small\pm0.004}$   | $0.020{\small\pm0.014}$ |
> | AutoEncoder for reconstruction | $0.099{\small\pm0.043}$​ | $0.152{\small\pm0.086}$​ | $0.045{\small\pm0.042}$​ |
>
> [cite1] PointNet: Deep Learning on Point Sets for 3D Classification and Segmentation. CVPR 2017
>
> [cite2] Yang et al. FoldingNet: Point Cloud Auto-encoder via Deep Grid Deformation. CVPR 2018
>
> ---
>
> Q2: Understanding of sentences and figures in this paper. “Minor: L187. is represented -> are Figure 3. (b) -> are these objects supposed to be at different angles? I cannot tell.”
>
> Q2.1: To help understand the sentence in “L187: To this end, the 3D structures encoded by a DNN is represented using the interaction between different 3D point cloud regions”, we answer the question what is the multi-order interaction.
>
> Q2.2: Are objects in Figure 3 (b) supposed to be at different angles?
>
> Q2.3: Whether the sentence in L187 and Figure 3 (b) are relevant?
>
> A: Thanks. For Q2.1, let us take a human face image as an example to introduce the multi-order interaction. To simplify the analysis, we set up a toy example, in which the face image only contains four elements of the *left-eye*, the *right-eye*, the *nose*, and the *mouth*. Then the interaction between the nose and the mouth can be represented as follows.
>
> $I(nose,mouth)=\frac{1}{3} \underbrace{ I(nose,mouth|S=\emptyset) }_{\rm interaction\ of\ order\ 0}$​
>
> $\qquad \qquad \qquad \qquad + \frac{1}{6}\underbrace{ I(nose,mouth|S=\\{left\textit{-}eye\\}) }_{\rm interaction\ of\ order\ 1} $
>
> $\qquad \qquad \qquad \qquad + \frac{1}{6}\underbrace{ I(nose,mouth|S=\\{right\textit{-}eye\\}) }_{\rm interaction\ of\ order\ 1}$
>
> $\qquad \qquad \qquad \qquad +\frac{1}{3} \underbrace{ I(nose,mouth|S=\\{left\textit{-}eye,right\textit{-}eye\\}) }_{\rm interaction\ of\ order\ 2},$​
>
> where $I(nose, mouth | S) = v(S∪\\{nose, mouth\\}) − v(S ∪\\{ nose \\}) − v(S ∪ \\{ mouth \\}) + v(S)$ (see Equation (6)).
>
> For Q2.2, objects in the ModelNet10 dataset and the ShapeNet dataset were placed with different rotation angles, but we rotate them to the same angle to simplify the comparison between the two objects at different angles. The first angle yields the maximum  attribution of region $i^*$​​, and the other angle yields the minimum attribution of region $i^*$​​​.
>
> For Q2.3, yes, Figure 3 (b) is a special case of the sentence in L187, because the regional attribution (*i.e.,* the Shapley value) visualized in Figure 3 (b) can be decomposed to the sum of multi-order interactions [25], as follows.
>
> $\phi(i) = \frac{1}{n} \sum_{m=0}^{n-1} \phi^{(m)}(i)$
>
> $ \phi^{(m)}(i) = \mathbb{E}\_{j \in N \setminus \\{ i \\}} [ \sum_{k=0}^{m-1} I^{(k)} (i,j) ]+\phi^{(0)}(i) $
> , where $\phi^{(0)}(i) \overset{\rm def}{=} v(\\{ i \\})-v(\emptyset)$
>
> ---
>
> Q3: Discussion about the prior work [cite3].
>
> A: Thanks. Yes, we agree with your insight that "PointNet++ is the network that pays most attention to intrinsic 3D structures: ‘masses,’ which can be why in many settings is still the better performing [cite3]." [cite3] found that different evaluation schemes, data augmentation strategies, and loss functions made a large difference in the classification of 3D point clouds. In contrast, our study analyzed the quality of knowledge representations of DNNs for 3D point cloud processing using metrics based on Shapley values and interactions. We will cite this paper in our final version and add discussions about its findings.
>
> [cite3] Goyal et al. Revisiting Point Cloud Shape Classification with a Simple and Effective Baseline

---

### Official Review · Reviewer_8Z6V · 2021-07-26

**Rating:** 7
**Confidence:** 3

**Summary:**

At a higher level, this paper is proposing several ways to measure the sensitivity and smoothness of representations learnt by 3D point cloud networks. This paper presents several ways to measure the sensitivity of point cloud networks to rotation, scale, translation and local structure. The Paper also proposes different metrics to evaluate the smoothness of encoded local 3d structures and the complexity of representations learnt by the network. The analysis is done using several point-based neural networks trained on the ModelNet40 classification task and shapenet part segmentation task.

**Limitations And Societal Impact:**

1. This is not a limitation per se, but it would be interesting to see how much data augmentation changes sensitivity to different perturbations.
2. Furthermore, there are several works on rotation invariant neural networks for 3d shapes. I wonder if the representations learnt by these architectures are different when evaluated using proposed metrics.


**Main Review:**

1. Paper is well written and claims are stated clearly.
2. The ideas explored in the paper are important for understanding how much point-based networks are vulnerable to different perturbations in the input and also how smooth/complex the learned representations are. This might lead to the design of better architecture that is robust to these perturbations.
3. Experiments are done on a variety of architectures and elucidate the ideas presented in the paper.

**Time Spent Reviewing:**

4.5

---

> ### Author Response · Authors · 2021-08-10
> **Response to Reviewer 8Z6V**
>
> Thank you for your great efforts on the review of this paper. We will try our best to answer all your questions.
>
>
>
> Q1: “This is not a limitation per se, but it would be interesting to see how much data augmentation changes sensitivity to different perturbations.”
>
> A: Thanks. We have followed your suggestions to **add experiments** to explore the effects of data augmentation on sensitivities, including (1) the translation data augmentation, (2) the scale data augmentation, (3) the rotation data augmentation around the y-axis, and (4) the rotation data augmentation around a random axis.
>
> We conducted experiments on two network architectures, *i.e.,* PointNet and DGCNN. As Table 1 shows, for each network, we trained five versions of this network with different data augmentation settings. We compared model A and B to explore effects of the translation data augmentation. We compared model A and C to explore effects of the scale data augmentation. We compared model A and D to explore effects of the rotation data augmentation around the y-axis. We compared model A and E to explore effects of the rotation data augmentation around a random axis. Each DNN was trained on the ModelNet10 dataset. The baseline network (version A) was trained following the standard and widely-used setting in [cite1]. All the other versions B-E were revised from such a standard setting to enable fair comparisons.
>
> **Table 1: Different versions of a specific network architecture using different data augmentation.**
>
> | Model                           | translation data augmentation | scale data augmentation | rotation data augmentation around the y-axis | rotation data augmentation around a random axis |
> | ------------------------------- | ----------------------------- | ----------------------- | -------------------------------------------- | ----------------------------------------------- |
> | A: baseline settings in [cite1] | $\checkmark$                  | $\checkmark$            | $\times$                                     | $\times$                                        |
> | B                               | $\times$                      | $\checkmark$            | $\times$                                     | $\times$                                        |
> | C                               | $\checkmark$                  | $\times$                | $\times$                                     | $\times$                                        |
> | D                               | $\checkmark$                  | $\checkmark$            | $\checkmark$                                 | $\times$                                        |
> | E                               | $\checkmark$                  | $\checkmark$            | $\times$                                     | $\checkmark$                                    |
>
> We found that the translation data augmentation decreased the translation sensitivity of a DNN (as Table 2 shows), the scale data augmentation decreased the scale sensitivity of a DNN (as Table 3 shows), the rotation data augmentation around the y-axis/a random axis decreased the rotation sensitivity of a DNN (as Tables 4 and 5 show).
>
> **Table 2: Translation sensitivity of DNNs with or without the translation data augmentation.**
>
> | Network architecture | Model A: w/ the translation data augmentation | Model B: w/o the translation data augmentation |
> | :------------------: | --------------------------------------------- | ---------------------------------------------- |
> |       PointNet       | $0.111{\small\pm0.053}$​                       | $0.160{\small\pm0.067}$​                        |
> |        DGCNN         | $0.048{\small\pm0.024}$​                       | $0.098{\small\pm0.054}$​                        |
>
> **Table 3: Scale sensitivity of DNNs with or without the scale data augmentation**
>
> | Network architecture | Model A: w/ the scale data augmentation | Model C: w/o the scale data augmentation |
> | :------------------: | --------------------------------------- | ---------------------------------------- |
> |       PointNet       | $0.025{\small\pm0.017}$​                 | $0.063{\small\pm0.047}$​                  |
> |        DGCNN         | $0.020{\small\pm0.015}$​                 | $0.028{\small\pm0.020}$​                  |
>
> **Table 4: Rotation sensitivity of DNNs w/ or w/o the rotation data augmentation around y-axis**
>
> | Network architecture | Model D: w/ the rotation data augmentation around y-axis | Model A: w/o the rotation data augmentation around y-axis |
> | :------------------: | -------------------------------------------------------- | --------------------------------------------------------- |
> |       PointNet       | $0.108{\small\pm0.047}$​​                                  | $0.155{\small\pm0.068}$                                   |
> |        DGCNN         | $0.158{\small\pm0.063}$                                  | $0.173{\small\pm0.072}$                                   |
>
> **Table 5: Rotation sensitivity of DNNs w/ or w/o the rotation data augmentation around a random axis**
>
> | Network architecture | Model E: w/ the rotation data augmentation around a random axis | Model A: w/o the rotation data augmentation around a random axis |
> | :------------------: | ------------------------------------------------------------ | ------------------------------------------------------------ |
> |       PointNet       | $0.057{\small\pm0.030}$                                      | $0.155{\small\pm0.068}$                                      |
> |        DGCNN         | $0.052{\small\pm0.021}$                                      | $0.173{\small\pm0.072}$                                      |
>
>
>
> Besides, we also compared the effects of rotation data augmentation and the effects of adversarial training on the rotation sensitivity. We conducted experiments on GCNN. We trained four versions of GCNN, including model A (see Table 1), model D (see Table 1), Model E (see Table 1), and the adversarially-trained GCNN.
>
> Table 6 shows that compared with the rotation data augmentation, the adversarial training based on rotations of point clouds had a greater impact on the rotation sensitivity. *I.e.,* the adversarially-trained DNN had lower rotation sensitivity than the DNN with the rotation data augmentation around the y-axis and the DNN with the rotation data augmentation around a random axis.
>
> **Table 6: Rotation sensitivity of DNNs with the rotation data augmentation and the adversarially-trained DNN**
>
> |                            Model                             | Rotation sensitivity    |
> | :----------------------------------------------------------: | ----------------------- |
> |        Model A: GCNN w/o  rotation data  augmentation        | $0.174{\small\pm0.066}$ |
> | Model D: GCNN w/  rotation data  augmentation around the y-axis | $0.147{\small\pm0.054}$ |
> | Model E: GCNN w/  rotation data  augmentation around a random axis | $0.048{\small\pm0.024}$ |
> |                **Adversarially-trained GCNN**                | $0.033{\small\pm0.011}$ |
>
> [cite1] DGCNN: Dynamic Graph CNN for Learning on Point Clouds. ACM Transactions on Graphics (TOG) 2019
>
>
> ---
>
> Q2: “Furthermore, there are several works on rotation invariant neural networks for 3d shapes. I wonder if the representations learnt by these architectures are different when evaluated using proposed metrics.”
>
> A: Thanks. We have followed your suggestions to **add experiments** to evaluate the representation quality of rotation-invariant DNNs using proposed metrics. We have conducted experiments on two rotation-invariant DNNs [cite2] (the network output is invariant, but intermediate-layer features are equivariant). Each rotation-invariant DNN was trained on the ModelNet10 dataset. We measured the rotation sensitivity on these two DNNs, as shown in Table 7. We found that compared with traditional DNNs (even the adversarially-trained GCNN) used in our paper, rotation-invariant DNNs were much more robust to the rotation.
>
> **Table 7: Comparison of sensitivities of DNNs used in our paper and rotation-invariant DNNs [cite2]**
>
> | DNNs  used in our paper        | rotation  sensitivity   |
> | ------------------------------ | ----------------------- |
> | PointNet                       | $0.159{\small\pm0.070}$​ |
> | PointNet++                     | $0.171{\small\pm0.064}$​ |
> | PointConv                      | $0.145{\small\pm0.060}$ |
> | DGCNN                          | $0.174{\small\pm0.075}$ |
> | GCNN                           | $0.174{\small\pm0.067}$ |
> | Adversarially-trained GCNN     | $0.034{\small\pm0.012}$ |
> | Rotation-invariant  PointNet++ | $0.002{\small\pm0.001}$​​ |
> | Rotation-invariant  DGCNN      | $0.010{\small\pm0.004}$​​​​ |
>
> [cite2] Shen et al. 3D-Rotation-Equivariant Quaternion Neural Networks. In ECCV 2020.

---

### Decision · Program_Chairs · 2021-09-27

**Decision:**

Accept (Poster)

**Comment:**

This paper introduces metrics that reflect the representation quality, and properties, acquired by a deep-net that is applied in 3D Point-Cloud (PC) processing tasks.
This paper presents several ways to measure the sensitivity of point cloud networks to rotation, scale, translation and local structure.
All reviewers laud the originality, clarity of the manuscript, and the ideas.
The initial paper lacked comparative analysis on different datasets, which the rebuttal seems to have addressed.
All reviewers agree that this paper provides an interesting result for the community to justify acceptance.